



Atmospheric
Chemistry
and Physics

# Attribution of Chemistry-Climate Model Initiative (CCMI) ozone radiative flux bias from satellites

**Le Kuai**[1,2], **Kevin W. Bowman**[1,2], **Kazuyuki Miyazaki**[1,3], **Makoto Deushi**[4], **Laura Revell**[5], **Eugene Rozanov**[6], **Fabien Paulot**[7], **Sarah Strode**[8], **Andrew Conley**[9], **Jean-François Lamarque**[9], **Patrick Jöckel**[10], **David A. Plummer**[11], **Luke D. Oman**[12], **Helen Worden**[9], **Susan Kulawik**[13], **David Paynter**[7], **Andrea Stenke**[14], and **Markus Kunze**[15]

[1]Jet Propulsion Laboratory, California Institute of Technology, Pasadena, CA, USA
[2]Joint Institute For Regional Earth System Science and Engineering, University of California, Los Angeles, CA, USA
[3]Japan Agency for Marine-Earth Science and Technology, Yokosuka, Kanagawa, Japan
[4]Meteorological Research Institute, Tsukuba, Ibaraki, Japan
[5]School of Physical and Chemical Sciences, University of Canterbury, Christchurch, New Zealand
[6]Physikalisch-Meteorologisches Observatorium Davos – World Radiation Center (PMOD/WRC), Davos, Switzerland
[7]NOAA, Geophysical Fluid Dynamics Laboratory, Princeton, NJ, USA
[8]USRA, NASA Goddard Space Flight Center, Greenbelt, MD, USA
[9]National Center for Atmospheric Research, Boulder, CO, USA
[10]Deutsches Zentrum für Luft- und Raumfahrt (DLR), Institut für Physik der Atmosphäre, Oberpfaffenhofen, Germany
[11]Climate Research Branch, Environment and Climate Change Canada, Montreal, Canada
[12]NASA Goddard Space Flight Center, Greenbelt, MD, USA
[13]Bay Area Environmental Research Institute, NASA Ames, Moffett Field, CA, USA
[14]Institute for Atmospheric and Climate Science, ETH Zürich (ETHZ), Zürich, Switzerland
[15]Freie Universität Berlin, Berlin, Germany

**Correspondence:** Le Kuai (lkuai@jpl.nasa.gov)

**Abstract.** The top-of-atmosphere (TOA) outgoing longwave flux over the 9.6 µm ozone band is a fundamental quantity for understanding chemistry–climate coupling. However, observed TOA fluxes are hard to estimate as they exhibit considerable variability in space and time that depend on the distributions of clouds, ozone ($O_3$), water vapor ($H_2O$), air temperature ($T_a$), and surface temperature ($T_s$). Benchmarking present-day fluxes and quantifying the relative influence of their drivers is the first step for estimating climate feedbacks from ozone radiative forcing and predicting radiative forcing evolution.

To that end, we constructed observational instantaneous radiative kernels (IRKs) under clear-sky conditions, representing the sensitivities of the TOA flux in the 9.6 µm ozone band to the vertical distribution of geophysical variables, including $O_3$, $H_2O$, $T_a$, and $T_s$ based upon the Aura Tropospheric Emission Spectrometer (TES) measurements. Applying these kernels to present-day simulations from the Chemistry-Climate Model Initiative (CCMI) project as compared to a 2006 reanalysis assimilating satellite observations, we show that the models have large differences in TOA flux, attributable to different geophysical variables. In particular, model simulations continue to diverge from observations in the tropics, as reported in previous studies of the Atmospheric Chemistry Climate Model Intercomparison Project (ACCMIP) simulations. The principal culprits are tropical middle and upper tropospheric ozone followed by tropical lower tropospheric $H_2O$. Five models out of the eight studied here have TOA flux biases exceeding 100 mW m$^{-2}$ attributable to tropospheric ozone bias. Another set of five models have flux biases over 50 mW m$^{-2}$ due to $H_2O$. On the other hand, $T_a$ radiative bias is negligible in all models (no more than 30 mW m$^{-2}$). We found that the atmospheric component (AM3) of the Geophysical Fluid Dynamics Labo-

ratory (GFDL) general circulation model and Canadian Middle Atmosphere Model (CMAM) have the lowest TOA flux biases globally but are a result of cancellation of opposite biases due to different processes. Overall, the multi-model ensemble mean bias is $-133 \pm 98 \, \text{mW m}^{-2}$, indicating that they are too atmospherically opaque due to trapping too much radiation in the atmosphere by overestimated tropical tropospheric $O_3$ and $H_2O$. Having too much $O_3$ and $H_2O$ in the troposphere would have different impacts on the sensitivity of TOA flux to $O_3$ and these competing effects add more uncertainties on the ozone radiative forcing. We find that the inter-model TOA outgoing longwave radiation (OLR) difference is well anti-correlated with their ozone band flux bias. This suggests that there is significant radiative compensation in the calculation of model outgoing longwave radiation.

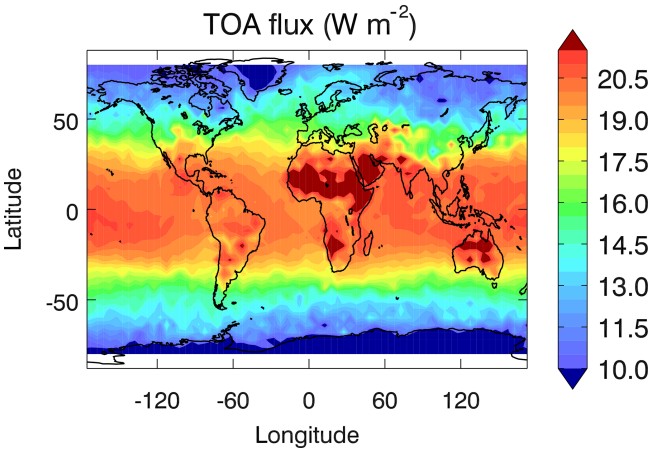

**Figure 1.** The clear-sky TES observed TOA flux at 9.6 µm $O_3$ band, annually averaged in 2006.

# 1   Introduction

Tropospheric ozone ($O_3$) is the third important anthropogenic greenhouse gas (GHG) in terms of radiative forcing (RF) as a consequence of $O_3$ precursor and methane ($CH_4$) emission increases from pre-industrial times to the present day. Tropospheric $O_3$ adjusted RF ranges widely from $+0.2$ to $+0.6 \, \text{W m}^{-2}$ computed from chemistry–climate model ensembles (IPCC AR5, 2013) (Bowman et al., 2013; Stevenson et al., 2013). The large uncertainty of the tropospheric $O_3$ RF is driven in part by the model responses to climate change. Without a good long-term record of the historical $O_3$ levels (Young et al., 2017; Gaudel et al., 2018), such estimates are highly dependent on the model assumptions of past $O_3$ levels. Differences between models in physical climate, chemical, and radiative processes conspire to complicate the assessment of the accuracy of these RF calculations. Consequently, a method to disentangle the key players caused the model differences to observations as well as the difference between the models is critical for robust estimates of chemistry–climate coupling.

About 80 % of tropospheric $O_3$ RF is due to $O_3$ longwave absorption, with the remaining 20 % from the shortwave absorption (IPCC AR5, 2013). In the longwave, 97 % of the total longwave absorption is in the 9.6 µm $O_3$ band (Rothman et al., 1987). The global outgoing longwave radiation (OLR) spectra were first observed from space for a few months in 1970. Radiance observations were taken during April 1970 and January 1971 by the NASA Infrared Interferometric Spectrometer (IRIS) and then from October 1997 for 9 months by the Interferometric Monitor of Greenhouse Gases (IMG) instrument, on board the Japanese Advanced Earth Observing Satellite "Midori" (ADEOS) satellite. Harries et al. (2001) showed that the changes in the greenhouse gas features between the observed spectra taken 30 years apart by these two instruments suggest increases in greenhouse gas forcing. Over the last two decades, a

new generation of thermal infrared satellite instruments has provided a unique opportunity to continuously monitor the outgoing radiances covering the 9.6 µm $O_3$ band globally, such as NASA's Tropospheric Emission Spectrometer (TES) and Atmospheric Infrared Sounder (AIRS), ESA's Infrared Atmospheric Sounding Interferometer (IASI), and NOAA's Cross-track Infrared Sounder (CrIS). These valuable long-term global measurements can be used to derive the top-of-atmosphere (TOA) $O_3$ band flux and the sensitivity of the flux to the vertical distributions of $O_3$, defined as instantaneous radiative kernels (IRKs) (Worden et al., 2011; Doniki et al., 2015).

The TES-observed global TOA outgoing fluxes at the 9.6 µm $O_3$ band in clear skies (Fig. 1) show strong geographic variations as a result of the short lifetime of $O_3$ (Worden et al., 2011; Bowman et al., 2013). Consequently, the global $O_3$ GHG effect is more unevenly distributed than long-lived GHGs, such as $CO_2$. In addition, the variations of the TOA fluxes are not only highly dependent on the distributions of $O_3$ but are also dependent on water vapor ($H_2O$), air temperature ($T_a$), and surface temperature ($T_s$) (Kuai et al., 2017).

There is an additional factor where the large-scale atmospheric structure sets the overall atmospheric opacity, which describes the fraction of the light that fails to pass through the atmosphere due to the absorption or scattering. For example, $O_3$ changes in more opaque regions, e.g., the western Pacific, a wet region due to convection, result in a much smaller change in TOA flux than in more transparent regions, e.g., the Middle East, a dry region due to downwelling (Kuai et al., 2017). This opacity has a direct impact on radiative forcing calculations.

Chemistry–climate models diverge significantly in the simulation of these processes, which are difficult to disentangle because it is hard to quantify the response of the TOA flux due to the change in atmospheric opacity. In this study,

we introduce a method to use observational-based IRKs to quantitatively estimate the contributions of the model biases in $O_3$, $H_2O$, $T_a$, and $T_s$ to the TOA flux biases.

The presence of clouds is the primary control on atmospheric opacity. Under the cloudy sky conditions, the roles of these variables other than clouds on TOA flux are much weaker. In addition, the variation in clouds could affect model estimates not only of the ozone but also of the flux sensitivity to ozone and other variables. Both ozone and sensitivity will impact the ozone radiative flux but in opposite directions. With cloud cover, the $O_3$ loss will be reduced. That means too many clouds would lead to more ozone production. The presence of the cloud would also cause weaker flux sensitivity to $O_3$ and other variables (IRKs). Therefore, the cloud effect is a battle between the impact on ozone estimation and the radiative sensitivity to ozone (IRK). The differences in cloud variations between the models will complicate the radiative effect. Furthermore, the study of the cloud effect is also currently limited by the global observations of total cloud cover and IRK product under realistic cloud conditions. Without knowing which models have better cloud cover, we benefit from using IRK based on the observed cloud-free data by TES. Therefore, here, we first try to access the role of $O_3$, $H_2O$, $T_a$, and $T_s$ in the variation of the TOA flux without cloud effect.

Worden et al. (2008) first attempted to disentangle these effects from satellites. They subsequently developed the IRK in Worden et al. (2011) for $O_3$, which is used in this study as a powerful tool to attribute model variability. IRKs for $O_3$ represent the sensitivity of TOA fluxes to the vertical distributions of the observed $O_3$. Aghedo et al. (2011) applied the TES IRKs to evaluate the $O_3$ radiative effect of chemistry–climate models' $O_3$ biases in the Atmospheric Chemistry Climate Model Intercomparison project (ACCMIP) (Lamarque et al., 2013). Bowman et al. (2013) found model OLR bias due to $O_3$ is correlated with RF in the ACCMIP models. This correlation helped to reduce the intermodel divergence in RF by about 30 % (Myhre et al., 2013). Doniki et al. (2015) updated the IRKs' calculation with a more accurate but computationally more complicated method, a five-angle Gaussian integration (GI) method, to replace the anisotropic approximation. They computed the $O_3$ IRKs with IASI observations and also showed that between the two methods there are about 20 % differences in IRKs and about 20 %–25 % differences in the longwave radiative effect (LWRE). They also found that the day and night difference of LWRE is mainly controlled by the $T_s$ change instead of $O_3$ amount change. Kuai et al. (2017) updated the computational method for the TES $O_3$ IRK product with the five GI method and revealed the hydrological controls on the global distribution of the $O_3$ GHG effect. The study showed that $H_2O$, $T_a$, and $T_s$ affect the $O_3$ IRK strength through relative humidity.

Therefore, the TOA flux in the 9.6 µm band depends on more than $O_3$. Consequently, in this study, we expand the TES observation-based IRKs to other quantities, including $H_2O$ profiles, $T_a$ profiles, and $T_s$. We apply these IRKs to help understand the reasons for the model divergence in the TOA flux.

The questions that have never been answered before include the following. (1) How do the model-based flux and the flux sensitivity compare to the observational-based flux and sensitivity? (2) How do they compare between the models? (3) How do the flux biases in models relate to the RF variation? Thus, benchmarking present-day $O_3$ band flux is the first step in answering all these questions and would help to further understand the correlations between the bias in TOA flux and the bias in $O_3$ RF, and eventually improve the estimation of the climate feedbacks from $O_3$ forcing.

To benchmark the model-simulated geophysical quantities, a recently developed multi-species multi-satellite Tropospheric Chemistry Reanalysis (TCR) product (Miyazaki et al., 2015) is used in this study to compare to the model results. This chemical reanalysis assimilates data from multiple satellites with sensitivity over complementary parts of the atmosphere, which provides better information than single-species chemical data assimilation. Satellite observations have the occasional issue of temporal discontinuity due to instrument performance and irregular spatial coverage, which can be circumvented by chemical data assimilation. Miyazaki et al. (2015) showed that statistically the model error against independent aircraft and ozonesonde observations in the assimilated species, e.g., $O_3$, $NO_2$, and CO, is significantly reduced. The multi-species assimilation improves the Northern/Southern Hemisphere OH ratio and provides the emission estimates with interannual variation. The comparison of $O_3$ reanalysis to the ACCMIP ensemble $O_3$ simulation in Miyazaki and Bowman (2017) quantified the model discrepancies in terms of seasonal amplitude, spatial variability, and interhemispheric gradient. For example, the ensemble mean is 6–11 ppb too high in the northern extratropics, while up to 18 ppb too low in the southern tropics over the Atlantic in the lower troposphere. In this study, we use the same $O_3$ reanalysis data (Miyazaki and Bowman, 2017) to understand the model bias in the CCMI project (Morgenstern et al., 2017), a follow-up model intercomparison study for ACCMIP. The multi-species assimilation also provides the opportunity to optimize the chemical-related species of $O_3$ and the emission sources of the precursors simultaneously. Further work by Miyazaki et al. (2017) showed that the surface emission of nitrogen oxides ($NO_x$) over a 10-year period (2005–2014) has a positive trend in regions including India, China, and the Middle East, but a negative trend over the US, southern Africa, and western Europe. The global total emission stays almost constant between 2005 (47.9 Tg N yr$^{-1}$) and 2014 (47.5 Tg N yr$^{-1}$). Therefore, the $O_3$ reanalysis data from TCR represent the state of the art for the current knowledge of the global distribution of tropospheric $O_3$ by combining the complementary information from model and satellite observations for $O_3$ and its precursors.

In this paper, we demonstrate a method to use the IRK products and the model biases relative to the reanalyzed tropospheric composition ($O_3$ and $H_2O$) and atmospheric state ($T_s$ and $T_a$) to quantitatively attribute the radiative biases of the flux in a suite of CCMI models to these dominant components. The method and IRKs are described in Sect. 2. The models and reanalysis data are introduced in the next section. Section 4 discusses the intercomparison between models' flux biases, the bias attribution to the dominant components, and the geospatial and vertical distribution of the biases. Lastly, the conclusion and future directions are summarized in Sect. 5.

## 2 Instantaneous radiative kernels (IRKs) for the climate variables

The TOA flux in the 9.6 μm $O_3$ band (Fig. 1) is defined as

$$F_{\text{TOA}} = \int\limits_{v} \int\limits_{0}^{2\pi} \int\limits_{0}^{\frac{\pi}{2}} L_{\text{TOA}}(v, \theta, \phi, q) \sin(\theta) \cos(\theta) \, \mathrm{d}\theta \mathrm{d}\phi \mathrm{d}v, \quad (1)$$

where $v$ is the frequency, integrated over the $O_3$ band from 980 to 1080 $\text{cm}^{-1}$. $L_{\text{TOA}}(v, \theta, \phi, q)$ is the upwelling TOA radiance at frequency $v$, zenith angle $\theta$, and azimuth angle $\phi$. We assume here that the radiance is symmetric in the azimuthal direction. The outgoing TOA radiances, $L_{\text{TOA}}$, are also a function of the atmospheric state, which is represented by variable "$q$", e.g., $H_2O$, $O_3$, and $T_a$, that is in turn a function of altitude, $z$.

The IRKs (Eq. 2) represent the sensitivities of the TOA radiative flux in the 9.6 μm $O_3$ band to the changes in the vertical distribution of an atmospheric variable.

$$\frac{\partial F_{\text{TOA}}}{\partial q(z_l)} = \int\limits_{v} \int\limits_{0}^{2\pi} \int\limits_{0}^{\frac{\pi}{2}} \frac{\partial L(v, \theta, \phi, q)}{\partial q(z_l)} \sin(\theta) \cos(\theta) \, \mathrm{d}\theta \mathrm{d}\phi \mathrm{d}v, \quad (2)$$

where $z_l$ is altitude in discretized level $l$. When $q$ represents the $T_s$, $z_l$ becomes a single surface value at $l = 0$. The partial derivative term on the right side of the equation is the spectral radiance Jacobians calculated analytically by the TES radiative transfer model.

In this study, we expanded the TES global $O_3$ IRKs to IRKs with respect to $H_2O$, $T_a$, and $T_s$. These TOA flux sensitivities still refer to the spectral window region in the 9.6 μm $O_3$ band for the flux. All the kernels are computed with the five-angle GI method (Doniki et al., 2015; Kuai et al., 2017). Figures 2a, c, and e show examples of IRK profiles for $O_3$, $H_2O$, and $T_a$ for 2006. The TOA flux is most sensitive to each variable at very different vertical levels. The $O_3$ IRK peaks in the middle and upper troposphere (600 to 200 hPa), a higher level than the peaks in both $H_2O$ and $T_a$ IRKs. The middle and upper tropospheric $O_3$ near 500 hPa has the largest impact on the TOA flux change (close to 1 mW m$^{-2}$ ppb$^{-1}$ in

the tropics). The $H_2O$ IRK peaks near 700 hPa, a little higher than the $T_a$ IRK. The $T_a$ IRK is maximal closest to the surface, suggesting that the $O_3$ band flux is most sensitive to boundary layer $T_a$ near 900 hPa. The strength of the peaks decreases with increasing latitude for all the three variables but the peak altitude does not change significantly except for the $H_2O$ IRKs in the polar region, which peaks at a slightly higher level than in lower latitudes.

In addition, the $T_s$ IRK is greater than zero, which means increases in $T_s$ would increase the outgoing TOA flux. However, the IRKs for the GHGs, i.e., $H_2O$ and $O_3$, are negative, because the increase in gas concentrations reduces the upwelling flux at TOA due to radiative absorption by the gas.

The global vertical distributions of the zonal averaged kernels for $O_3$, $H_2O$, and $T_a$ are also shown below their profile plots in Fig. 2b, d, and f. The sensitivities of the TOA flux to these three variables are strongest in the tropics and decrease with latitude. Furthermore, the IRK for $T_s$ is also shown in Fig. 2g and h. Unlike the other IRKs, the $T_s$ IRK is not a function of altitude, so we show the winter (December–February) and summer (June–August) seasonal average of its global distribution. The flux sensitivities are found to be largest over the major deserts, like the Sahara, the Middle East, and Australia, corresponding to the regions with the highest values of $T_s$. We also notice that the values of the $T_s$ IRKs in the Intertropical Convergence Zone (ITCZ) are much lower than those in the subtropics, which suggests that the atmosphere opacity has an impact on the strength of the $T_s$ IRKs.

## 3 A method to attribute the flux biases

The flux biases between observations and models under the clear-sky conditions can be described as

$$\delta F_{\text{TOA}}^i \approx \sum_{l \in L} \frac{\partial F_{\text{TOA}}^i}{\partial O_3^{i,l}} \delta O_3^{i,l} + \sum_{l \in L} \frac{\partial F_{\text{TOA}}^i}{\partial H_2O^{i,l}} \delta H_2O^{i,l}$$
$$+ \sum_{l \in L} \frac{\partial F_{\text{TOA}}^i}{\partial T_a^{i,l}} \delta T_a^{i,l} + \sum_{l \in L} \frac{\partial F_{\text{TOA}}^i}{\partial T_s^{i,l}} \delta T_s^{i,l}, \quad (3)$$

where $\partial F_{\text{TOA}}^i$, is the total TOA flux bias in the $O_3$ band at the $i$th location. The four terms on the right-hand side of the equation are the products of the IRKs and the biases in the geophysical quantities (i.e., $O_3$, $H_2O$, $T_a$, and $T_s$). These biases are then vertically integrated on index $l$, over domain $L$, which in our case is the troposphere. The summation is the vertical integral from the surface to the tropopause.

Here, we assume that the biases due to other physical processes, e.g., surface emissivity or other atmospheric species, have much less influence on the TOA flux variation. For example, the model bias in global emissivity is not accessible but is believed to be quite small compared to $O_3$, $H_2O$, $T_a$, and $T_s$. We also assume that the nonlinearity terms are much smaller than these four first-order terms.

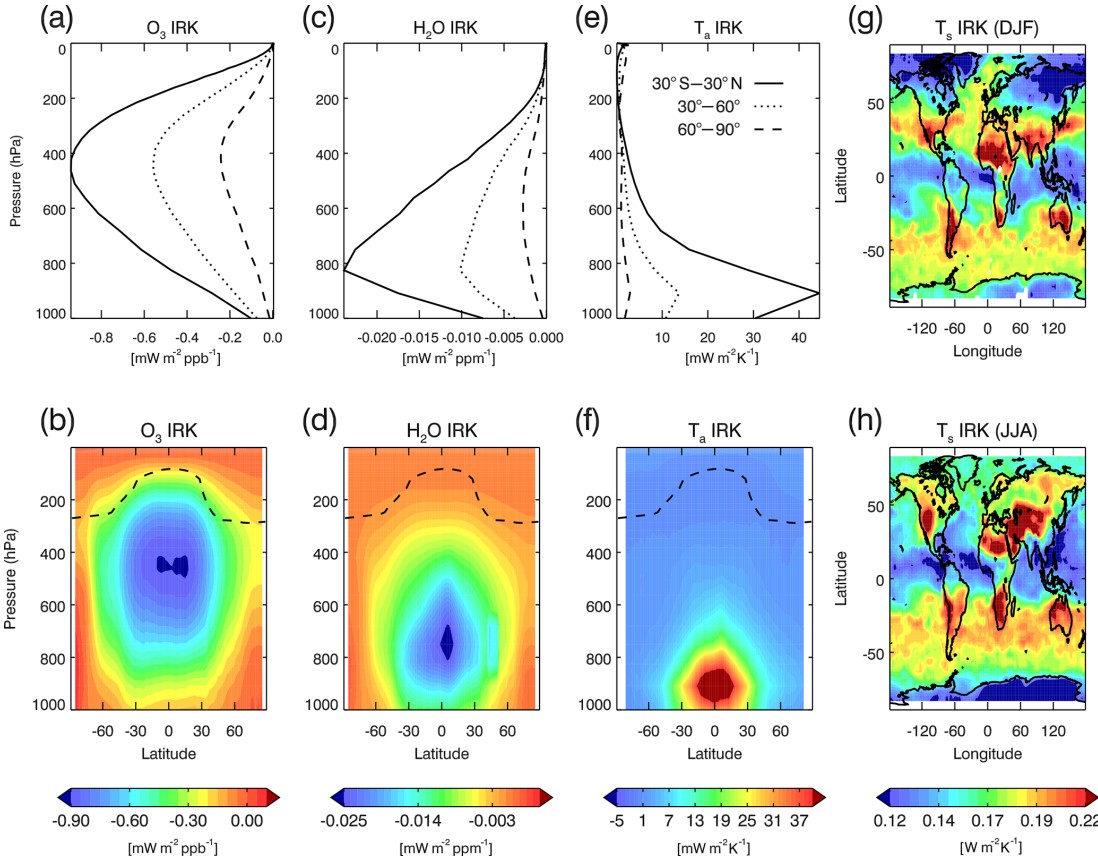

**Figure 2.** TES 2006 IRK for four primary components ($O_3$, $H_2O$, $T_a$, and $T_s$). Panels **(a)**, **(c)**, and **(e)** are IRKs of latitudinal band averages in the tropics (30° S–30° N), midlatitudes of both hemispheres (30–60°), and high latitudes of both hemispheres (60–90°). The figures below them are the pole-to-pole vertical distribution of the zonally averaged IRK. The global distribution of IRK for $T_s$ is plotted for winter season (December to February) **(g)** and summer season (June to August) **(h)**.

Following Bowman et al. (2013), the delta terms in Eq. (3) are the model biases with respect to the reanalysis data, defined as below:

$$\delta q^{i,l} = q^{i,l}_{\text{mod}} - q^{i,l}_{\text{assim}}, \tag{4}$$

where $q^{i,l}_{\text{mod}}$ and $q^{i,l}_{\text{assim}}$ represent the model and reanalysis $O_3$, $H_2O$, $T_a$, or $T_s$ at the $i$th location and the $l$th altitude level, respectively.

The mean flux bias or the mean bias components from tropospheric uncertainties are calculated from Eqs. (3) and (4) as

$$\delta F^j_q = \frac{1}{N_j} \sum_{i \in D_j} \sum_{l \in L} w_i \frac{\partial F^i_{\text{TOA}}}{\partial q^{i,l}} (q^{i,l}_{\text{mod}} - q^{i,l}_{\text{assim}}), \tag{5}$$

where $w_i$ is area weighted for the latitude bands, $D_j$ is a set of observed locations, $N_j$ is the number of locations in the domain of $D_j$ and tropospheric levels of $L$ up to the tropopause. We use the chemical tropopause $O_3 = 150\,\text{ppb}$ (Naik et al., 2005; Hansen et al., 2007; Bowman et al., 2013; Kuai et al., 2017). The domain of $D_j$ can be zonal bands for

the zonal mean or global area for the global mean, respectively. The global mean of the flux bias and its components will be denoted as $\overline{\delta F}$ and $\overline{\delta F_q}$, respectively.

## 4 Chemistry–climate models and the reanalysis data

### 4.1 Models and simulations

We analyze six models from the CCMI study (Table 1) (Hegglin and Lamarque, 2015; Morgenstern et al., 2017; Eyring et al., 2013). It is a combined activity of the International Global Atmospheric Chemistry (IGAC) and Stratosphere-troposphere Processes And their Role in Climate (SPARC) (Randel et al., 2004). The CCMI coordinates a number of model experiments that capture the variability and evolution of air quality, tropospheric chemistry, stratospheric $O_3$, and global climate. This approach builds on the legacy of previous chemistry–climate model intercomparisons, such as the Chemistry-Climate Model Validation (CCMVal, Morgenstern et al., 2010; SPARC, 2010, Eyring et al., 2010) and the ACCMIP. In this study, we use the experiment REF-C1,

which is analogous to the REF-B1 experiment of CCMVal-2 (Table S30 in Morgenstern et al., 2017). REF-C1 requires using historic forcing and observed sea surface conditions. The models are free-running and simulate the recent past (1960–2010). We did not choose to use REF-C1SD (specified dynamics) because specified dynamics nudged the wind and temperature of the model to be constrained to the reanalysis data. The long-term climatological biases relative to the reanalysis between the models are minimized. Our study aims to find a correlation between the present-day radiative bias and the RF from present day to future by the model predictions. Therefore, we prefer to keep the model differences in simulating longer-term climatology between their free runs.

We note that SOCOL3 and EMAC are both based on different versions of the ECHAM5 climate model. We also added two additional model simulations with AM3 from NOAA and CESM from NCAR. These two simulations are not the specific CCMI experiment run; however, these two models have been used in many studies, and including them in this study provides more useful information on the TOA flux diversity among the most recent models.

### 4.2 Tropospheric Chemistry Reanalysis (TCR-1) data

We computed the biases in the geophysical variables between the model and the reanalysis data. To compute the $O_3$ bias in models, we used the satellite-based $O_3$ reanalysis from multi-constituent multi-satellite data assimilation: Tropospheric Chemistry Reanalysis version 1 (TCR-1) (Miyazaki et al., 2015; Miyazaki and Bowman, 2017) as the best synthesis of the observations. The reanalysis provides comprehensive spatiotemporal and multi-variable evaluation of model performance that complements direct comparisons against individual measurements, which may suffer from significant sampling bias (Miyazaki and Bowman, 2017).

TCR-1 assimilated multiple species data from multiple satellite products for the period from 2005 to 2017, e.g., combined TES and Microwave Limb Sounder (MLS) observations for $O_3$, integrated Ozone Monitoring Instrument (OMI), Scanning Imaging Absorption Spectrometer for Atmospheric Chartography (SCIAMACHY), and Global Ozone Monitoring Experiment-2 (GOME-2) for tropospheric $NO_2$ column, Measurements Of Pollution In The Troposphere (MOPITT) for CO, and MLS for $HNO_3$. TCR-1 used a global chemistry–transport (CTM) Model for Interdisciplinary Research on Climate with chemistry (MIROC-Chem; Watanabe et al., 2011) as a forecast, which includes 92 species and 262 reactions. The model has 2.8° horizontal resolution with 32 vertical layers up to 4 hPa. The data assimilation was based on an ensemble Kalman filter with 32 ensemble members, which was used to simultaneously optimize concentrations and emissions of various species.

As summarized by Miyazaki and Bowman (2017), the mean bias in the reanalysis dataset against the World Ozone and Ultraviolet Data Centre (WOUDC) ozonesonde obser-

vations is from $-3.9$ to $-2.9$ ppb at the NH high latitudes (55–90° N); $-0.9$ to $-0.1$ ppb at the NH midlatitudes (15–55° N); and $-1.0$ to $-0.1$ ppb at the SH midlatitudes (55–15° S), between 850 and 500 hPa. On average, the bias is about 0.9 ppb at the tropics and midlatitudes between 500 and 200 hPa. These biases are much smaller than biases in the model simulation without data assimilation, demonstrating that the multi-satellite data assimilation provides comprehensive constraints on the entire tropospheric profile of $O_3$.

For the purpose of consistency, we also use outputs of $H_2O$, $T_a$, and $T_s$ from the reanalysis to estimate the model biases. In the reanalysis calculation, meteorological fields simulated by the atmospheric general circulation model MIROC-AGCM (Watanabe et al., 2011) were nudged toward the 6-hourly ERA-Interim meteorological reanalysis (Dee et al., 2011) for zonal wind ($\tau = 1$ d) and temperature ($\tau = 3$ d) to reproduce past meteorological fields while simulating short-term ($< 6$ h) meteorological variations, which were used to drive the CTM, as similarly employed in CCMI C1SD simulations. Thus, the reanalysis dataset provides realistic and comprehensive estimates for both chemical and meteorological fields required for the TOA flux evaluations.

## 5 Results

### 5.1 The latitudinal distribution of the TOA flux bias

Figure 3 shows the latitudinal distribution of the zonal and annual mean of the TOA flux bias from each model relative to the reanalysis. The largest divergence between the models is located at the tropics where most models underestimate the flux, with the exception of CMAM. The low bias in the model ensemble implies the model atmosphere is more opaque than the chemical reanalysis, leading to a 133 mW m$^{-2}$ outgoing flux reduction on average. The TOA flux in an opaque atmosphere is less sensitive to the changes in tropospheric composition than a more transparent one. Under those conditions, the models would underestimate the radiative feedback from composition since the IRKs estimated under an opaque atmosphere will be weaker than those under a realistic (more transparent) atmosphere.

Two models that have larger low biases at the equatorial region than other models are SOCOL3 and MRI-ESM1r1. Their global means of the flux bias are more than $-200$ mW m$^{-2}$ (Table 2). The following analysis will help to clarify the source of the bias in the models.

### 5.2 Flux bias attribution

The total TOA flux bias is caused by biases from atmospheric composition and temperature. In order to determine the primary drivers of these biases, we apply the IRKs to the differences between model and the chemical reanalysis as described in Eq. (3). Figure 4 shows the contribution of $O_3$

**Table 1.** The chemistry–climate models and their experiment simulations.

|  | Model | Institutes | CCMI runs | | |
|---|---|---|---|---|---|
| 1 | CMAM | CCCma, Environment and Climate Change Canada | REF-C1 | r1i1p1 | v1 |
| 2 | SOCOL3 | ETH-Zurich, PMOD/WRC | REF-C1 | r1i1p1 | v1 |
| 3 | GEOSCCM | NASA/GSFC | REF-C1 | r1i1p1 | v1 |
| 4 | EMAC-L47MA | DLR-IPA, KIT-IMK-ASF, KIT-SCC-SLC, FZJ-IEK-7, FUB, UMZ-IPA, MPIC, CYI | REF-C1 | r1i1p1 | v1 |
| 5 | EMAC-L90MA | | REF-C1 | r1i1p1 | v1 |
| 6 | MRI-ESM1r1 | MRI | REF-C1 | r1i1p1 | v1 |
| 7 | AM3 | NOAA GFDL | – | – | – |
| 8 | CESM | NCAR | – | – | – |
|  | ERA-Interim, TCR-1 | Reanalysis | – | – | – |

**Table 2.** The global mean of the flux bias (mW m$^{-2}$) and the dominant components due to tropospheric $O_3$, $H_2O$, $T_a$, and $T_s$. The numbers in parentheses are the standard deviation of the zonal distribution. For the ensemble, the standard deviation is computed from the variation between the models.

| Models | $\overline{\delta F}$ | $\overline{\delta F_{T_s}}$ | $\overline{\delta F_{T_a}}$ | $\overline{\delta F_{H_2O}}$ | $\overline{\delta F_{O_3}}$ |
|---|---|---|---|---|---|
| AM3 | −78.1 (46.2)[a] | −11.7 (30.0) | −14.2 (7.8) | 87.7 (76.4)[b] | −140 (117.1)[b] |
| SOCOL3 | −283.6 (290.0)[b] | −11.6 (22.3) | 5.9 (13.9) | −94.5 (124.6)[b] | −183.5 (187.5)[b] |
| GEOSCCM | −139.9 (112.0) | −27.2 (38.5) | 1.1 (12.0) | −80.3 (120.9)[b] | −33.5 (28.0) |
| CMAM | 42.9 (118.5)[a] | −100.2 (93.3)[b] | 22.5 (40.5) | 127.9 (100.7)[b] | −7.3 (28.8) |
| EMAC-L47MA | −154.4 (150.0) | −3.3 (27.3) | 2.9 (14.8) | −28 (56.1) | −125.9 (128.2)[b] |
| EMAC-L90MA | −130.2 (142.7) | −8.7 (36.0) | −4.1 (6.6) | −30.5 (63.0) | −86.9 (108.7)[b] |
| MRI-ESM1r1 | −228.5 (281.8)[b] | −2.7 (17.9) | −0.1 (6.8) | −43.6 (86.6) | −182 (213.7)[b] |
| CESM | −91.1 (89.0) | −31.1 (33.9) | 6.4 (13.9) | −69.7 (113.0)[b] | 3.3 (43.8) |
| Ensemble | −132.9 (98) | −24.6 (32) | 2.6 (10) | −16.4 (81) | −94.5 (75) |

[a] The models that have relative small global and annual averaged TOA flux bias. [b] The extreme values for the large biases.

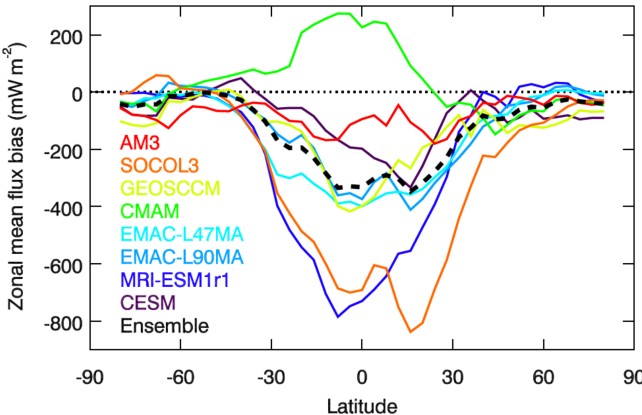

**Figure 3.** The latitudinal distribution of the zonal flux bias (model – reanalysis) with latitude weight.

(blue), $H_2O$ (green), $T_s$ (red), and $T_a$ (yellow) for each model to the total TOA flux bias (black). The global mean bias is summarized in Table 2.

In general, $O_3$ and $H_2O$ are the two dominant drivers for most models where the large biases are concentrated in the tropics and subtropics. There are only three models (GEOSCCM, CMAM, and CESM) whose $O_3$ radiative biases ($\overline{\delta F_{O_3}}$ in Table 2) are less than 50 mW m$^{-2}$ and are almost negligible zonally. While the flux bias is better represented in these models, it does not follow that they represent tropospheric $O_3$ more accurately, as will be shown in the following section. The other five models (AM3, SOCOL3, EMAC-L47MA, EMAC-L90MA, and MRI-ESM1r1) have significant negative peaks at low latitudes (Figs. 3 and 4), actually resulting from their strong $O_3$ contributed biases (from 80 to 180 mW m$^{-2}$; numbers are footnoted as b in Table 2).

The TOA flux bias from $H_2O$ is the second largest component for most models. Similar to $O_3$, most models show the fluxes are biased low in the tropics due to the $H_2O$ uncertainties with the exception of CMAM, which has the strongest global mean bias (127.9 mW m$^{-2}$). Note that, in the reanalysis, no data assimilation (or nudging) was applied for specific humidity. Watanabe et al. (2011) demonstrated a dry bias in the lower troposphere and a wet bias in the middle and upper troposphere in MIROC-AGCM, primarily attributable to temperature biases. Nevertheless, the reported $H_2O$ biases can be greatly reduced in the reanalysis because of the nudging applied for temperature.

The flux bias due to $T_a$ is found to be negligible in all models, which indicates that the model $T_a$ estimates provide rea-

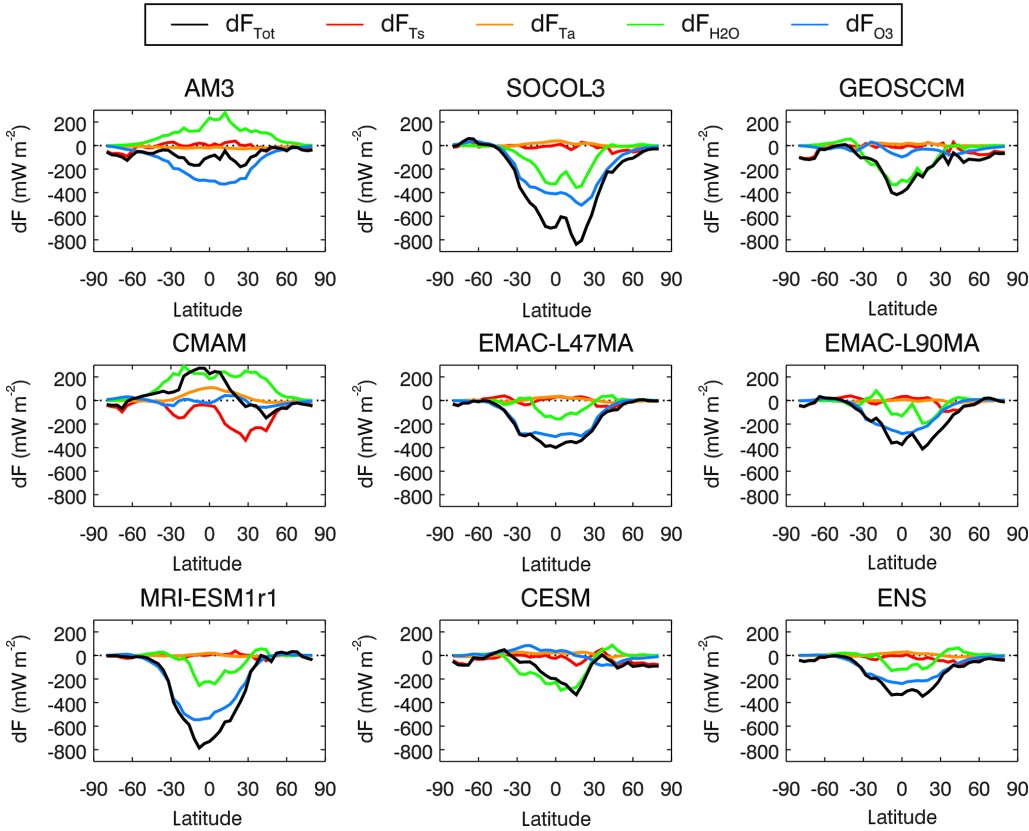

**Figure 4.** The attribution of the total TOA flux bias for each model to four dominant components and their latitudinal distribution. The black curves are the same as the colored curves in Fig. 3.

sonable radiative fluxes. $T_s$ radiative bias is also meridionally weak relative to the flux bias in $O_3$ and $H_2O$ (Fig. 4). With the exception of CMAM, the $T_s$ ensemble global mean bias is less than 35 mW m$^{-2}$ (see Table 2). Figure 4 suggests the strong bias from $T_s$ in CMAM ($-100.2$ mW m$^{-2}$) comes from the two subtropical regions.

Interestingly, the positive flux bias due to $H_2O$ ($127.9$ mW m$^{-2}$) is compensated by the negative flux bias due to $T_s$ ($-100.2$ mW m$^{-2}$) in CMAM, leading to the lowest global mean in $\overline{\delta F}$ ($42.9$ mW m$^{-2}$, calculated with Eq. 5). This compensation is also true for AM3 but between a positive $H_2O$ radiative bias ($87.7$ mW m$^{-2}$) and negative $O_3$ radiative bias ($-140$ mW m$^{-2}$). This analysis reveals that these two models are both right but for wrong – and opposite – reasons.

However, all the other models have a strong negative global mean bias and are mostly driven by the two major components ($O_3$ and $H_2O$). SOCOL3 and MRI-ESM1r1 are the two models that have the strongest low bias up to $-200$ mW m$^{-2}$, which is mainly due to their strong $O_3$ radiative bias ($-180$ mW m$^{-2}$). Their $O_3$ estimates are both biased high in the tropics and subtropics. We will show later that such bias is particularly strong in the upper troposphere.

## 5.3 Vertically resolved radiative bias of the $O_3$, $H_2O$, and $T$

The zonal flux biases among the models are both significant and mainly in the tropics. However, those biases are the vertically integrated product of the model profile bias and the IRKs both with their own vertical structures. The vertically resolved radiative bias can provide more insight into the processes leading to the biases. To further investigate, we examined the vertically resolved flux bias for $O_3$, $H_2O$, and $T_a$ (Figs. 5–7) and the global distribution for $T_s$ (Fig. 8). These are computed from Eq. (3) before the vertical summation. These figures show that the maximum contribution to the flux bias is a balance between the peak of the IRKs (Fig. 1) and the peak of the geophysical quantities' bias (Figs. 9–12). The positive tropical $O_3$ radiative bias for GEOSCCM, CMAM, and CESM is commonly centered in the midtroposphere, corresponding to the peak of the IRKs (Fig. 5). On the other hand, the primary $O_3$ flux bias contribution in the tropics for SOCOL3, EMAC-L47MA, EMAC-L90MA, and MRI-ESM1r1 is in the upper troposphere around 200 hPa even though the IRKs are roughly half the peak sensitivity. These strong negative biases exceed 15 mW m$^{-2}$.

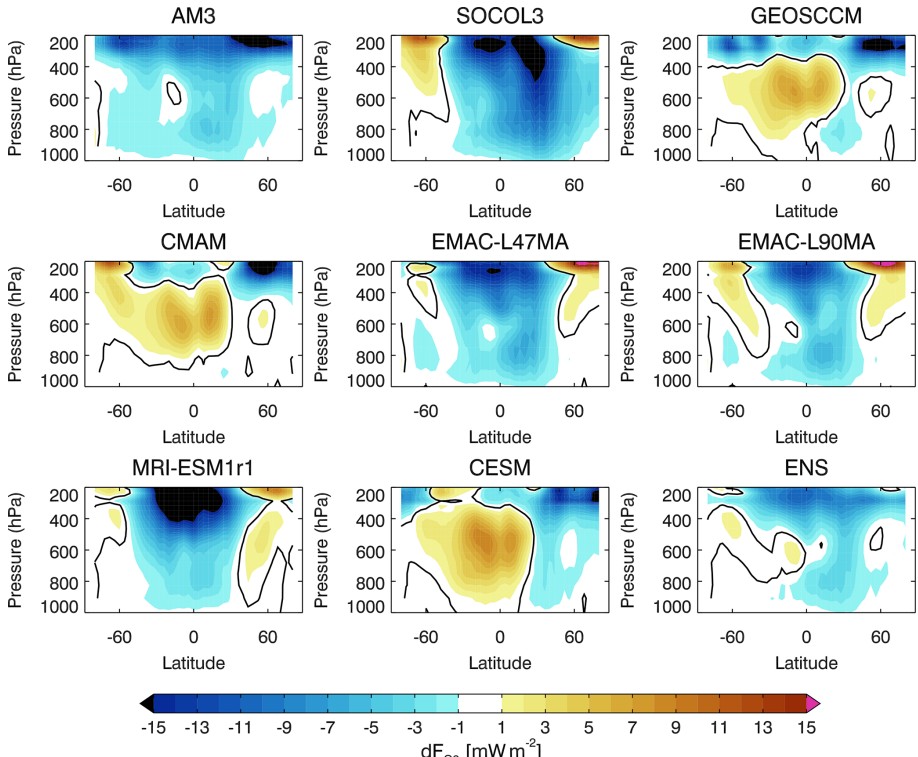

**Figure 5.** Vertical resolved $O_3$ radiative bias. The black curves are the zero lines.

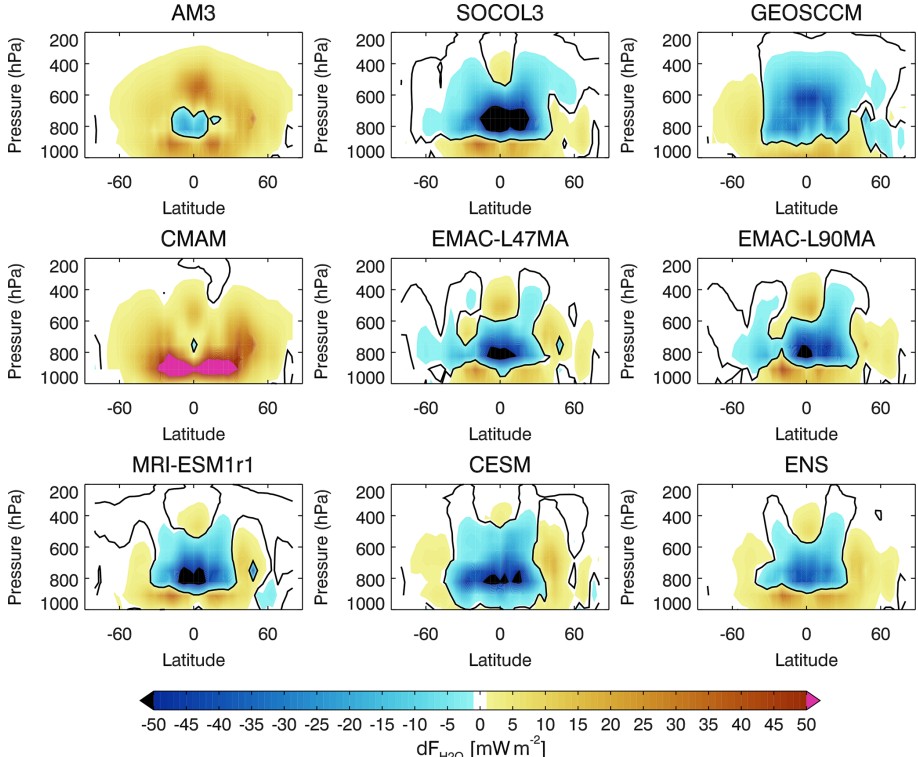

**Figure 6.** Vertical resolved $H_2O$ radiative bias. The black curves are the zero lines.

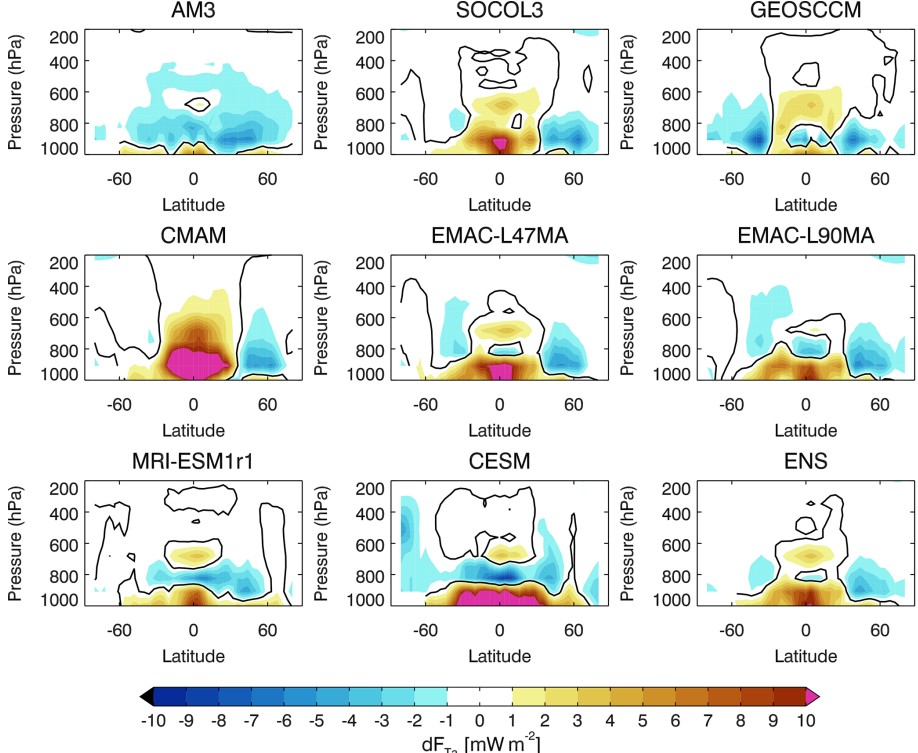

**Figure 7.** Vertical resolved $T_a$ radiative bias. The black curves are the zero lines.

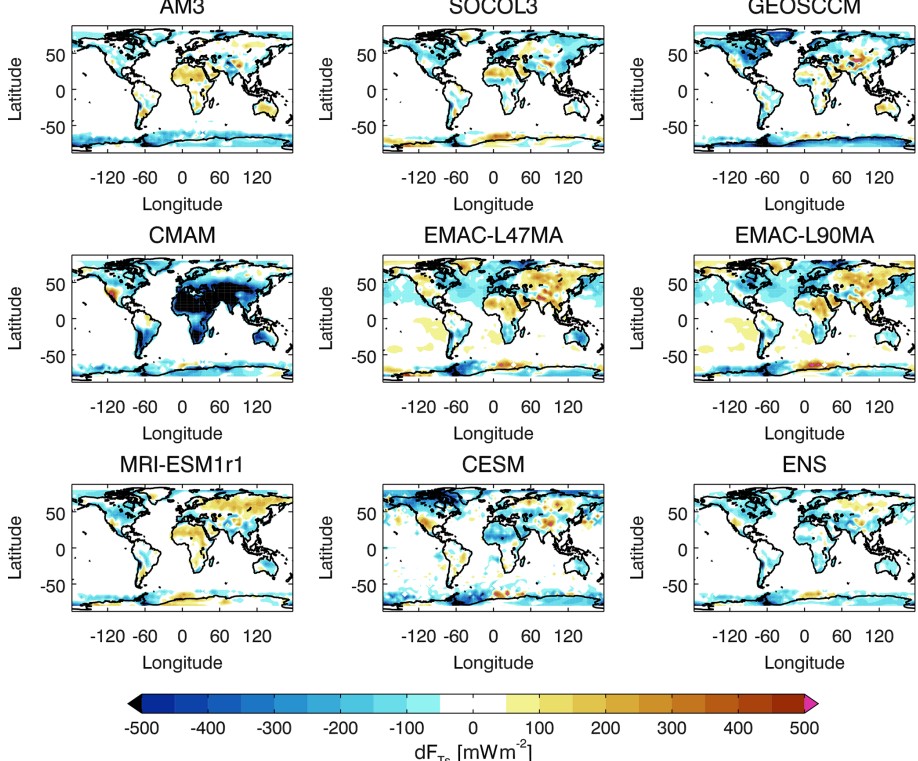

**Figure 8.** Global distribution of the $T_s$ radiative bias.

The strong tropical $H_2O$ radiative bias collapses to shallower tropical regions below 400 hPa and is maximized near 800 hPa for most models exceeding $50\,mW\,m^{-2}$ (Fig. 6). CMAM has the unique and strongest net positive bias of above $50\,mW\,m^{-2}$ centered lower and close to 900 hPa. While most models' flux bias is centered near 800 hPa, particularly GEOSCCM, AM3, and CESM show a more vertically uniform – and opposing – flux bias.

Figure 7 indicates that tropical $T_a$ radiative bias is largely negligible for vertical layers above 600 hPa as a consequence of the rapid decrease in sensitivity of the $T_a$ IRKs. The maximum bias is in the lower troposphere between 900 hPa and the surface. CMAM and CESM both show the strongest positive bias exceeding $10\,mW\,m^{-2}$ over most of the tropics. However, CESM has a compensating negative bias from 700 to 800 hPa that leads to a mean global bias of only $6.4\,mW\,m^{-2}$ (Table 2), whereas CMAM has a positive bias throughout, leading to an atmospheric $T_a$ radiative bias of $22.5\,mW\,m^{-2}$, the largest of the models studied here.

Surprisingly, the model ensemble $T_s$ turns out to be the second largest contributor to the total bias (Table 2), instead of $H_2O$, driven primarily by three models: CMAM, CESM, and GEOSCCM, as shown in Fig. 8. CMAM shows a negative bias that covers all of Africa, exceeding $500\,mW\,m^{-2}$, and Asia centered over India. Consequently, CMAM has the largest total bias ($-100.2 \pm 93.3\,mW\,m^{-2}$). CESM and GEOSCCM $T_s$ radiative biases, on the other hand, are centered at high latitudes in the western hemisphere over the eastern US and Canada, exceeding $300\,mW\,m^{-2}$.

The vertically and spatially concentrated radiative biases provide clues as to what processes are the most important for the total flux bias. These processes drive the distribution of the constituents, which we will discuss in detail in the next sections.

## 5.4 The spatial source of TOA flux bias

The source of the attributed flux biases can be traced back to their spatial origins, which can provide more insight into the underlying processes and the differences between the models.

### 5.4.1 $O_3$ bias

Figure 9 shows a vertically resolved zonal averaged distribution of $O_3$ biases between the model and the chemical reanalysis similar to that in Fig. 5. Three models (GEOSCCM, CMAM, and CESM) have the weakest globally averaged $O_3$ radiative bias reported in Table 2 ($-33.5$, $-7.3$, and $3.3\,mW\,m^{-2}$). These three models also have the lowest $O_3$ bias in tropical troposphere on average ($-1.1\,ppb$, $-1.3\,ppb$, and $3.0\,ppb$, reported in Table 3) and as a consequence have weaker radiative bias in the region with the strongest $O_3$ IRK globally TS1 (0.9, 1.4, and $2.1\,mW\,m^{-2}$; reported in Table 4). On the other hand, the global $O_3$ bias is greater than 7 ppb for

**Table 3.** Models' bias in the tropical troposphere between 25° S and 25° N, and below 200 hPa.

| Models | $\overline{\delta O_3}$ (ppb) | $\overline{\delta H_2O}$ (ppm) | $\overline{\delta T_a}$ (K) |
|---|---|---|---|
| AM3 | 7.4 (5.0) | −586.5 (669.9) | −1.0 (0.8) |
| SOCOL3 | 13.4 (6.8) | 330.3 (1067.3) | 0.1 (0.4) |
| GEOSCCM | −1.1 (4.3) | −506.6 (949.1) | 0.5 (0.5) |
| CMAM | −1.3 (3.9) | 18.1 (949.5) | 1.0 (0.4) |
| EMAC-L47MA | 10.4 (6.1) | −20.2 (729.1) | −0.2 (0.6) |
| EMAC-L90MA | 8.2 (5.6) | 203.1 (845.1) | −0.9 (1.0) |
| MRI-ESM1r1 | 13.7 (9.3) | 386.9 (679.9) | 0.1 (0.4) |
| CESM | −3.0 (4.4) | 22.0 (630.7) | 0.3 (0.7) |

**Table 4.** Models' flux bias ($mW\,m^{-2}$) in the tropic troposphere between 25° S and 25° N, and below 200 hPa.

| Model | $\overline{\delta F_{O_3}}$ | $\overline{\delta F_{H_2O}}$ | $\overline{\delta F_{T_a}}$ |
|---|---|---|---|
| AM3 | −4.8 (3.7) | 6.9 (10.5) | −0.7 (1.4) |
| SOCOL3 | −9.2 (5.3) | −9.1 (21.6) | 1.0 (2.2) |
| GEOSCCM | 1.0 (3.3) | −8.5 (14.1) | 0.7 (1.4) |
| CMAM | 1.4 (2.8) | 9.1 (18.5) | 3.1 (5.0) |
| EMAC-L47MA | −7.2 (4.8) | −3.2 (15.3) | 1.0 (2.9) |
| EMAC-L90MA | −5.6 (4.2) | −2.7 (16.4) | 0.2 (2.2) |
| MRI-ESM1r1 | −10.0 (7.4) | −5.5 (16.4) | 0.3 (2.0) |
| CESM | 2.1 (3.4) | −8.3 (15.0) | 1.3 (4.5) |
| Ensemble | −4.06 (4.90) | −2.66 (7.02) | 0.86 (1.1) |

all the other models and results in a large $O_3$ radiative bias in the tropics, especially SOCOL3 (13.4 ppb in the tropical $O_3$ bias and $-9.2\,mW\,m^{-2}$; see Tables 3 and 4) and MRI-ESM1r1 (13.7 ppb and $-10\,mW\,m^{-2}$).

GEOSCCM, CMAM, and CESM commonly have a vertically compensated pattern in the tropics that is biased high in the upper troposphere while biased low in the middle and lower troposphere (Fig. 9). Their $O_3$ low biases in the middle troposphere are approximately 5 to 10 ppb, where the peak of the IRK centered, but the high biases in the upper troposphere are about 5 ppb. Such a high–low pattern leads to compensation during the vertical integration through the troposphere into the radiative effect at the top of the atmosphere. The corresponding vertical resolved $O_3$ radiative bias for these three models in Fig. 5 shows the consistent tropical vertical distribution but in an opposite sign since the $O_3$ IRK is negative. In contrast, the other five models have vertically systematic biases high in the tropical $O_3$, and the biases increase from the middle troposphere to the upper troposphere. Especially SOCOL3 and MRI-ESM1r1 strongly overestimate $O_3$ by more than 20 ppb in a wide region of tropical upper troposphere. The $O_3$ radiative biases in this region remain significantly high, stronger than $-15\,mW\,m^{-2}$, causing these two models to have the highest $O_3$ radiative biases in the global and annual mean (both about $-183\,mW\,m^{-2}$).

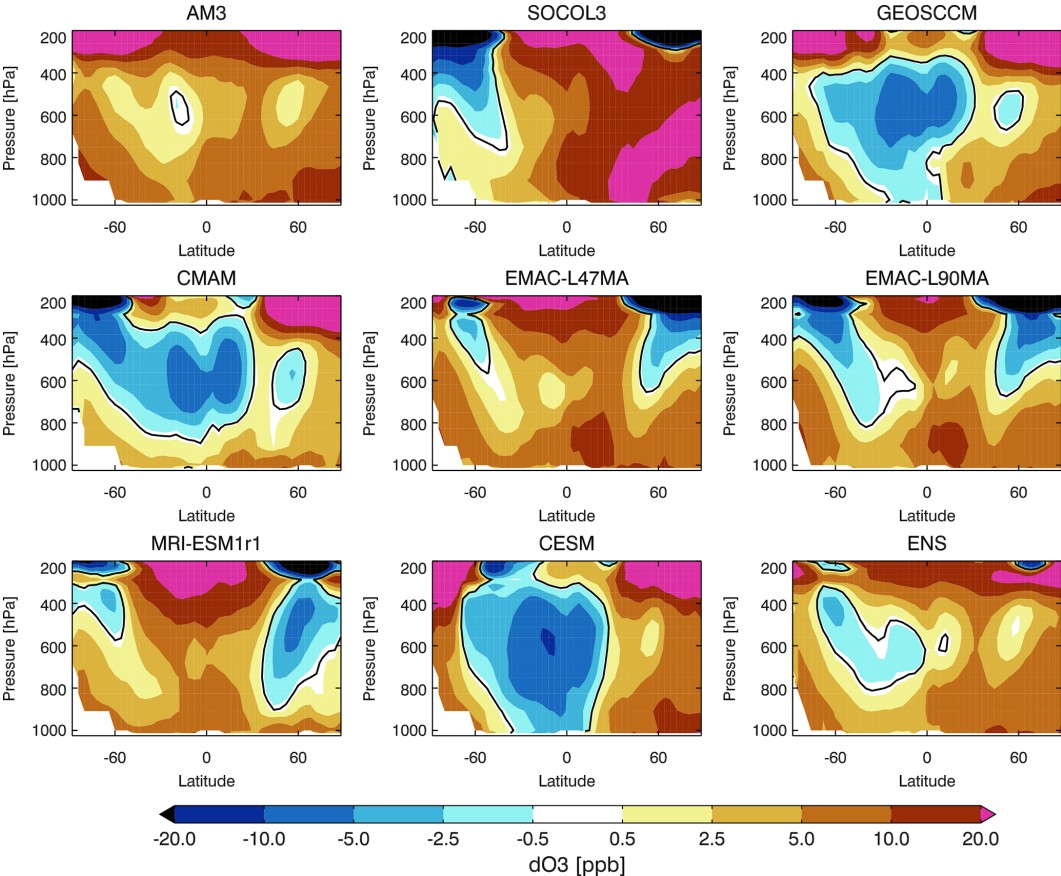

**Figure 9.** The zonal averaged vertical–latitudinal distribution of $O_3$ model biases to the TCR-1 $O_3$ assimilation data. The black curves are the zero lines.

The systematic bias in the entire tropical tropospheric $O_3$ and strong overestimation of upper troposphere in SOCOL3 and MRI-ESM1r1 could be caused by several factors. For example, the transport from the lower stratosphere could be too high. Alternatively, precursor emissions of tropospheric $O_3$ could also be too high. The analysis with the spatially explicit biases provides important clues to implicate the specific processes that individual modeling groups can investigate.

The GEOSCCM has been used to study the tropospheric $O_3$ response to variations in the El Niño–Southern Oscillation (ENSO), where Oman et al. (2011, 2013) compared the model to satellite observations. These regular comparisons may have led to the improved simulation of tropospheric $O_3$ profiles and consequently lower vertical $O_3$ bias. The GEOSCCM model in the CCMI study uses the tropospheric–stratospheric chemical package developed within the Global Modeling Initiative (GMI) program (Duncan et al., 2007), which has more realistic ozone chemistry, an internally generated quasi-biennial oscillation, an improved air–sea roughness parameterization and other improvements (Oman and Douglass, 2014).

Nielsen et al. (2017) showed that GEOSCCM has successfully reproduced the changes in the quasi-global (60° S–60° N) annual mean trend in total $O_3$ column since 1960s to the present day. For the present-day atmosphere, simulated tropospheric partial column $O_3$ from GESCCM Ref-C1 for CCMI was compared to satellite observations of OMI and MLS (Ziemke et al., 2011). The differences are mostly a few Brewer–Dobson units (DU) except in the Northern Hemisphere subtropics and middle latitudes in autumn and winter with the 4–6 DU biases which are under investigation.

The finding that SOCOL3 and MRI-EMS1r1 both have strong overestimates in the tropical upper troposphere is also understandable. SOCOL3 is the third generation of the coupled chemistry–climate model (CCM) SOCOL (modeling tools for studies of SOlar Climate Ozone Links). Several steps have been taken to improve the SOCOL model simulation of $O_3$. Stenke et al. (2013) first attempted to reduce the $O_3$ bias in their middle atmosphere by updating their middle-atmosphere general circulation with an advanced advection scheme. Revell et al. (2015) revealed that ozone precursor emissions are the biggest players that control the global mean change in tropospheric ozone. In a parallel study, Revell et al. (2018) developed an updated version of "SOCOL3.0", "SOCOL3.1", to reduce the tropospheric ozone bias. By improving the treatment of ozone sink processes, the tropo-

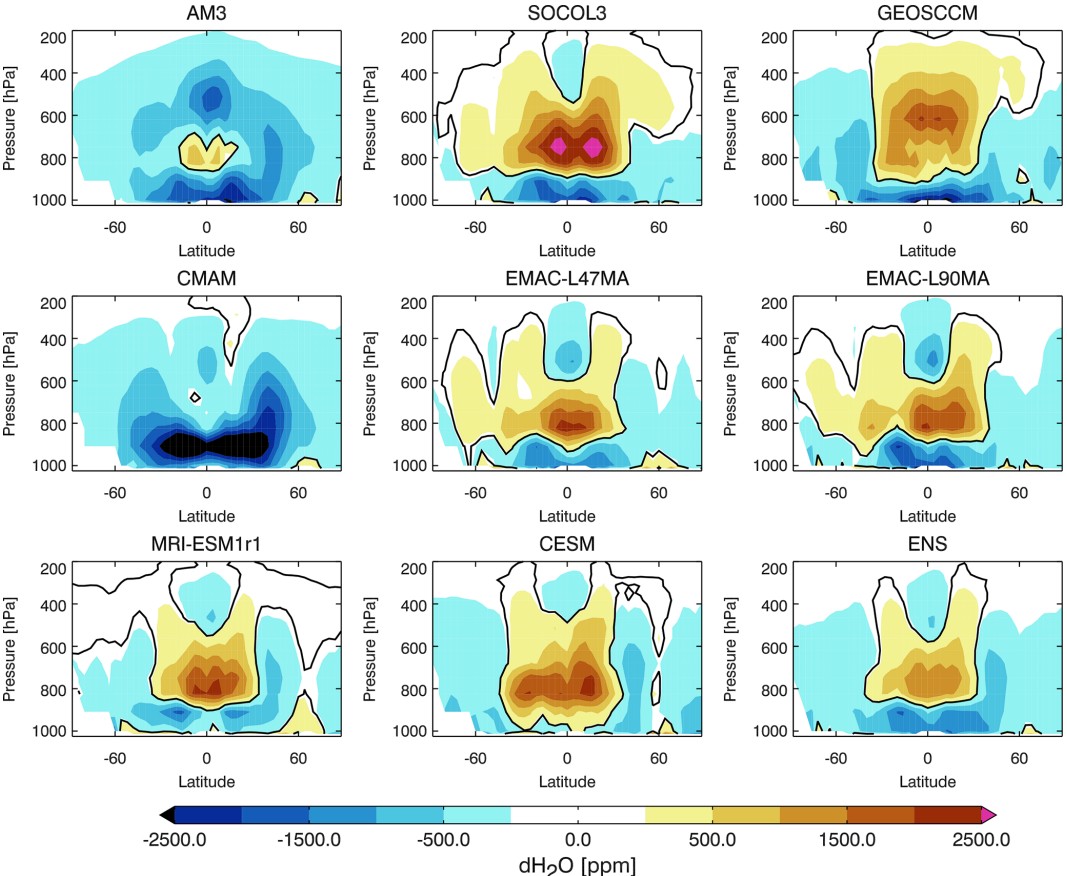

**Figure 10.** The zonal averaged vertical–latitudinal distribution of $H_2O$ biases (model to the ERAI reanalysis data). The black curves are the zero lines.

spheric column ozone bias in "SOCOLv3.1" is reduced up to 8 DU, mostly due to the inclusion of $N_2O_5$ hydrolysis on tropospheric aerosols. We expect that the future similar analysis with the SOCOL3.1 could show a reduced flux bias for this model.

Meanwhile, the strong tropical upper tropospheric $O_3$ biases in MRI-EMS1r1 are believed to be related to the weak tropical convective updraft and the large lightning $NO_x$ emissions in the model. The model with weak updraft fails to bring enough low $O_3$ air from the surface to the upper troposphere in the tropics or overestimates the upper tropospheric mixing of stratospheric ozone-rich air. In addition, the global lightning $NO_x$ ($LNO_x$) emission used in MRI-EMS1r1 is $10\,\mathrm{TgN}\,\mathrm{yr}^{-1}$. The best estimate of annual mean $LNO_x$ based on satellite data assimilation is $6.3\,\mathrm{TgN}\,\mathrm{yr}^{-1}$ (Miyazaki et al., 2014). The $LNO_x$ in GEOSCCM is approximately $5\,\mathrm{TgN}\,\mathrm{yr}^{-1}$ (Martini et al., 2011), which shows less tropical upper tropospheric $O_3$ bias compared to MRI-EMS1r1. Thus, the overestimation of the $O_3$ precursor in the upper troposphere is another reason for too much $O_3$. Figure A1 shows the improvement in the radiative biases due to less $O_3$ bias in the experiment by half the $LNO_x$ emissions in MRI-EMS1r1 (see the Appendix).

In summary, the potential reasons for the prevalence of $O_3$ radiative bias in the tropical middle and upper troposphere in the models could be due to following facts: (1) the tropical $O_3$ IRK is strongest in this region (Fig. 2); (2) the largest $O_3$ bias in the models also centered in the same place (e.g., SOCOL3 and MRI-EMS1r1; Fig. 9); (3) the simulations with the systematic bias throughout the tropical troposphere, when vertically integrated, accumulated into a larger column bias when compared to the models with vertically random biases.

### 5.4.2 $H_2O$ bias

$H_2O$ turns out to be the primary contributor for three models (GEOSCCM, CMAM, and CESM) since their $O_3$ radiative bias is small. It is also the second dominant driver after $O_3$ in the other five models. Different from $O_3$, $H_2O$ IRKs in Fig. 2 show the strongest sensitivity to the tropical lower troposphere centered at 800 hPa, where $H_2O$ is most concentrated globally. We found the model biases in $H_2O$ are strongest in the tropical lower troposphere. It explains why the strongest radiative bias from $H_2O$ is also located in the tropical region near 800 hPa in all models as shown in Fig. 6. Figure 10 and

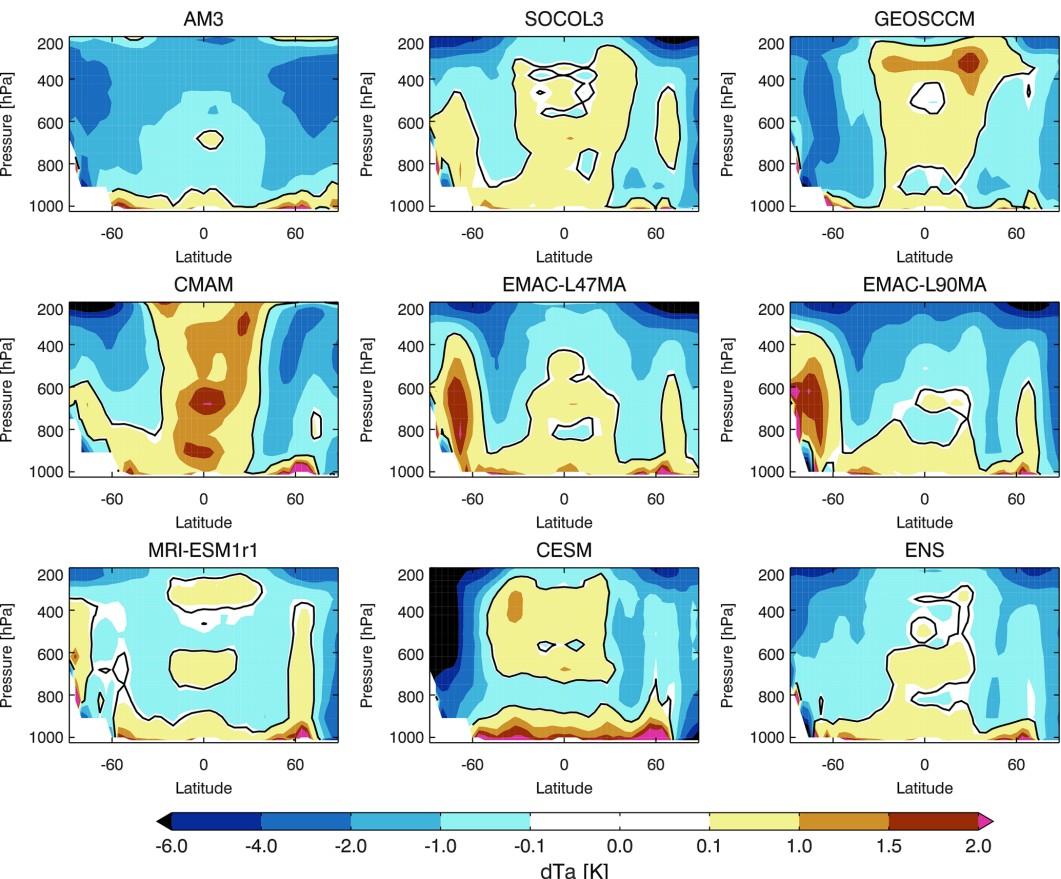

**Figure 11.** The zonal averaged vertical–latitudinal distribution of $T_a$ biases from models to the reanalysis data. The black curves are the zero lines.

Table 3 further help to indicate that $H_2O$ is biased low only in two models, AM3 ($-586.5$ ppm) and CMAM ($-506.6$ ppm). We note that $H_2O$ IRKs are also negative as $O_3$. Therefore, these two models have the unique overestimates in $H_2O$ radiative bias at low latitudes (see Fig. 4), while all the other models are predominantly biased high in tropical $H_2O$ concentrations, which result in the negative radiative biases.

### 5.4.3 $T_a$ bias

We found the $T_a$ radiative biases in these model ensembles are all negligible. There are two reasons. One is that the $T_a$ biases are small overall (less than 2 K) even at the tropical lower troposphere (below 1 K on average in Table 3). The other reason is that the compensation in the vertical integration helps to reduce the radiative bias at the top of atmosphere.

Figure 11 shows that the model biases in $T_a$ range within $\pm 2$ K for all the models because the current chemistry–climate models have been well developed to simulate the global atmospheric $T_a$ fields relative to reanalysis. The region with strongest sensitivity, identified by the $T_a$ IRKs (Fig. 2), is the tropical lower troposphere (the region within $\pm 30°$

and below 800 hPa). The $T_a$ biases in the tropics shift between positive and negative vertically in most models except CMAM, which is systematically biased (Fig. 11). The oscillated $T_a$ biases suggest that simulated air temperatures stay around the reanalyzed profiles. These models better represent the air temperature than the trace gases like $H_2O$ and $O_3$. The oscillation around the reanalyzed profile leads to vertical compensation in the air $T_a$ radiative bias. Therefore, the flux bias from $T_a$ is a small component compared to the radiative bias from $O_3$ and $H_2O$. Figure 4 suggests the only model that has a small tropical peak in the $T_a$ radiative component is CMAM, which has the strongest $T_a$ radiative bias ($22.5 \pm 40.5$ mW m$^{-2}$ in Table 2) among all the models. Figure 11 shows that this model has a deep region with a strong bias of about 2 K in tropical areas and also persistently overestimated $T_a$ vertically. Figure 7 further suggests the strong radiative bias in CMAM mainly comes from the tropical lower troposphere ($> 10$ mW m$^{-2}$ below 800 hPa). The other models have vertical compensation in the tropics (less than 0.5 K bias on average; see Table 3), and therefore they are less biased in TOA flux (less than 1 mW m$^{-2}$ in Table 4). The two EMAC models both have strong biases at the southern high latitudes but still have a weak radiative effect

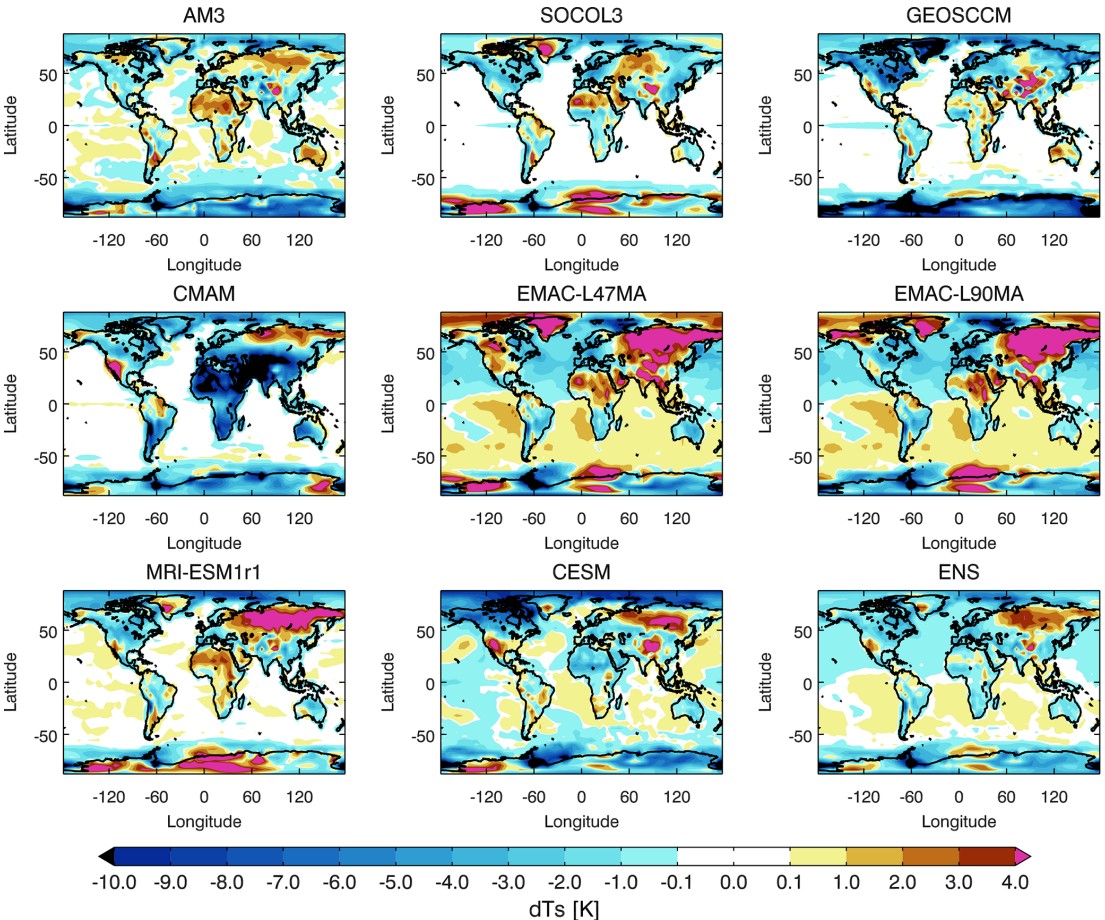

**Figure 12.** The global distribution of $T_a$ biases from models to the reanalysis data.

in this region (see Fig. 7) due to much weaker $T_a$ IRK at high latitudes.

### 5.4.4  $T_s$ bias

The global distribution of the $T_s$ biases indicates that the biases in sea surface temperature (SST) are smaller than the biases in land $T_s$ for all the models (Fig. 12) because the CCMI experiment (REF-C1) selected in this study used the observed SST. CMAM is the model that has the strongest $T_s$ radiative bias ($-100.2 \pm 93.3$ mW m$^{-2}$ in Table 2), which peaks in both subtropical regions (Fig. 4). These large negative biases are due to the large underestimates of the $T_s$ over the major deserts, e.g., the Sahara, the Middle East, and Australia (Fig. 12). In other words, the real deserts' surface is hotter than the model's prediction. At the same time, the $T_s$ IRKs at the subtropical desert region are also strongest globally, since the TOA flux is more sensitive to $T_a$ when the atmosphere is transparent, which is due to the downdraft of the Hadley cell controlling the region (Kuai et al., 2017). The downwelling airflow results in less precipitation and less clouds, as well as higher $T_s$ during summer over the desert

surface. These factors cause the CMAM to have the largest $T_s$ radiative bias compared to all the other models.

In contrast, two EMAC models and MRI-ESM1r1 also have strong high bias in $T_s$ in Siberia ($> 4$ K in Fig. 12), but the radiative bias is much less significant compared with the Middle East in CMAM in Fig. 8. The $T_s$ IRKs are weaker during the winter season in high latitudes than the low latitudes if the $T_s$ is low (Fig. 2). However, the IRKs at the subtropical desert region stay strong during winter. Therefore, the annual mean of the $T_s$ radiative bias is much weaker in Siberia in two EMAC models and MRI-ESM1r1 than the Middle East region in CMAM. Consequently, the global annual means of the $T_a$ radiative biases for two EMAC models and MRI-ESM1r1 are small, although the large biases in $T_s$ are found in their Siberian region.

## 6  Correlation to the broadband OLR

The analysis up to this point has been limited to the 9.6 μm band. We posed the question as to whether biases in this band could provide any insight into biases in the entire OLR band. To that end, we found an anti-correlation ($R = -0.6$) be-

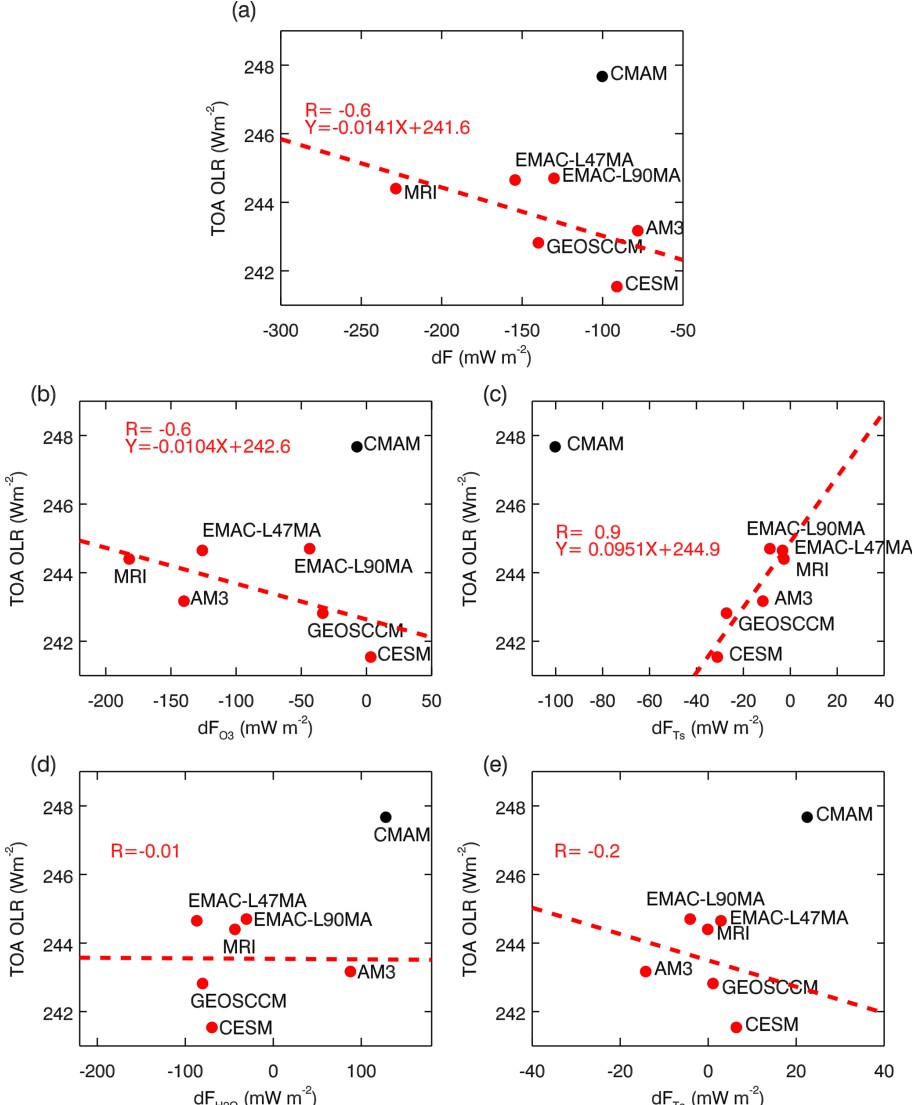

**Figure 13.** The correlation of the ozone band TOA flux biases to the model-calculated broadband OLR **(a)** and the correlation of the attributed radiative components to the broadband OLR **(b–e)**.

tween the global mean of the $O_3$ band flux biases and the clear-sky broadband OLR calculated internally by the models, as shown in Fig. 13a. The CMAM OLR is inconsistent with the ensemble (more than $2\,\mathrm{W\,m^{-2}}$ higher than all the other models), and therefore it is excluded in the correlation. Interestingly, a very similar regression line and anti-correlation coefficient ($R = -0.6$) are found between the $O_3$ radiative bias and the broadband OLR (Fig. 13b). The similar regression line indicates that the $O_3$ radiative bias dominates the $9.6\,\mu\mathrm{m}$ TOA flux distribution, which is confirmed by the attribution analysis that $O_3$ radiative bias is the largest term in five of eight models. The anti-correlation suggests a radiative compensation between the $9.6\,\mu\mathrm{m}$ band and the other parts of the OLR, assuming a constant globally integrated OLR at TOA. More interestingly, a strong correlation

($R = 0.9$) is found between $T_\mathrm{s}$ radiative bias and broadband OLR (Fig. 13c) because the $T_\mathrm{s}$ affects the entire baseline of the outgoing radiance and its radiative effect plays the same role in the $O_3$ band as in the entire OLR. However, there is no significant correlation found between $T_\mathrm{a}$ radiative bias and OLR, likely because there is no coherent bias in $T_\mathrm{a}$ radiative effect between the $O_3$ band and in the entire OLR. There is neither a correlation between the $H_2O$ radiative bias and broadband OLR. $H_2O$ absorption is ubiquitous in the OLR. Consequently, biases in the $9.6\,\mu\mathrm{m}$ band do not drive the magnitude of the overall $H_2O$ absorption in spite of the $H_2O$ biases.

The anti-correlation between the biases in the $9.6\,\mu\mathrm{m}$ band and in the entire OLR band would suggest some bias drivers in the $9.6\,\mu\mathrm{m}$ band must play different roles at the other part

of the OLR band. The further investigation of these processes would help to explain the radiative effect of different biases on the OLR estimations from models (Huang et al., 2008, 2014).

## 7   Conclusions

We have demonstrated a new method to quantitatively attribute the biases in $O_3$ band TOA flux from chemistry–climate model ensembles to $O_3$, $H_2O$, $T_a$, and $T_s$ radiative components without cloud effect using observationally constrained IRKs in the clear sky. The study also provides the first vertically and globally resolved view of the radiative bias for each component. An IRK depicts the sensitivity of TOA fluxes to the vertical distribution of the geophysical quantities, such as $O_3$, $H_2O$, $T_a$, and $T_s$. While the products of 9.6 µm $O_3$ band IRK for $O_3$, $H_2O$, $T_a$, and $T_s$ have been developed with the satellite observations by Aura TES, the record could be extended by MetOp-IASI and SNPP-CrIS Fourier Transform Spectrometer (FTS) measurements. We compute the model biases against reanalysis data for four key variables: $O_3$, $H_2O$, $T_a$, and $T_s$. Especially for $O_3$ biases, the newly developed TCR-1 $O_3$ assimilation data (Miyazaki et al., 2015; Miyazaki and Bowman, 2017) are, for the first time, used as the state-of-the-art benchmark for tropospheric $O_3$ in models. These specific bias comparisons for the CCMI study cause the modelers to investigate the reasons for these biases and motivate them to improve their simulations. For example, MRI-ESM1r1 shows the reduced $LNO_x$ emission help to improve their tropical upper tropospheric $O_3$ and its radiative bias.

$O_3$ abundance is found to be the dominant driver for the ensemble flux bias. Tropical tropospheric $O_3$ is too high for most models and accounts for about 70 % of the flux bias (Table 2). The second driver in the model ensemble becomes the $T_s$ instead of $H_2O$ because the $T_s$ radiative components are commonly biased low in the model ensemble, while the $H_2O$ radiative biases between models are biased randomly in both directions with large diversity. For individual models, however, $H_2O$ is the second most important driver, a larger component than $T_s$, for many cases such as AM3, SOCOL3, and MRI-ESM1r1.

In addition to determining that the tropospheric $O_3$ and $H_2O$ are overestimated, and the surface is too cold, the study also tells us the geolocations, in latitudes and altitudes, of the deviations in these geophysical quantities that propagate into the flux bias.

The largest spread of the flux bias between the models is found in the tropics. The principal contributors governing each model are different and controlled by different processes over different regions. The flux biases in five of the eight models (AM3, SOCOL3, EMAC-L47MA, EMAC-L90MA, and MRI-ESM1r1) are primarily driven by too much $O_3$ in the tropical middle and upper troposphere. $H_2O$

is a big driver in five models (AM3, SOCOL3, GEOSCCM, CMAM, and CESM). $T_s$ is an important contributor in CMAM in addition to its $H_2O$.

Although AM3 and CMAM overall have relative lower TOA flux biases globally, we found they are actually right for the wrong reasons. In AM3, the dominant positive $H_2O$ radiative bias (87.7 mW m$^{-2}$ in Table 2) happens to be canceled by the dominant negative $O_3$ component ($-140$ mW m$^{-2}$), while in CMAM, the large positive $H_2O$ component (127.9 mW m$^{-2}$) is mostly being compensated by $T_s$ radiative bias ($-100.2$ mW m$^{-2}$). The two relatively young models among the model ensembles, SOCOL3 and MRI-ESM1r1, have a large potential to be improved for their fluxes by reducing their strong negative radiative biases from both tropical upper tropospheric $O_3$ and tropical lower tropospheric $H_2O$.

On average, the model ensemble underestimates the flux by about 133 mW m$^{-2}$ due to overestimated tropical tropospheric $O_3$ and $H_2O$. The underestimate of the TOA flux implies the model atmosphere is too opaque. In a more opaque atmosphere, the change in flux will be weaker for the same change in tropospheric $O_3$ because the sensitivity (i.e., IRKs) is weaker. With such feedback, the $O_3$ RF, the changes in $O_3$ GHG effect from pre-industrial times to the present day would likely be underestimated. The opacity of the atmosphere is controlled by climate processes, such as the hydrological cycle, that is shown can indirectly affect the $O_3$ GHG effect and RF, as discussed in Kuai et al. (2017).

The spatially explicit and process-focused differences could be used as a basis for emergent constraints (Bowman et al., 2013). New techniques such as hierarchical emergent constraints (HECs) can harness this spatial information so that specific processes affecting $O_3$ RF can be identified (Bowman et al., 2018). Moreover, if this correlation exists between the TOA flux bias and the $O_3$ RF, then a similar issue could be found in the RF of other GHGs, such as $CO_2$ and $CH_4$. That is a subject for future research.

Finally, although the chemical reanalysis dataset provides comprehensive information on model radiative biases, we need to understand its performance. For instance, further improvements are still needed for lower tropospheric $O_3$ (Miyazaki and Bowman, 2017). Ingesting more datasets and applying a bias correction procedure would be useful to improve reanalysis accuracy. The lower tropospheric $O_3$ analysis would benefit from the recently developed satellite retrievals with high sensitivity to the lower troposphere (Fu et al., 2018) and the optimization of additional precursor emissions.

## Appendix A

Here, we compare the MRI-EMS1r1 experiment run RefC1_50 % LNO$_x$ with its RefC1 run. The new run's emission decreases by about 50 % compared with the original run. The global lightning NO$_x$ emission annual mean in 2006 simulated in the experiment run is reduced from $\sim$ 10.79 TgN yr$^{-1}$ in RefC1 to $\sim$ 5.21 TgN yr$^{-1}$. The 10-year average changes from $\sim$ 10.44 to $\sim$ 5.18 TgN yr$^{-1}$.

We found the total flux bias is much reduced due to the improved O$_3$ radiative bias (Fig. A1a, b). As we expected, the vertical resolved O$_3$ radiative bias shows that the overestimation of the tropical upper tropospheric O$_3$ radiative bias is much weaker in the new run (Fig. A1c, d). This improvement is due to the lower O$_3$ biases in this region caused by reduced LNO$_x$ emission (Fig. A1e, f).

We also see some changes in the latitudinal distributions of the H$_2$O radiative bias. This is because the reduction of the upper tropospheric O$_3$ will cause the model responses in the O$_3$ heating rate, which would have radiative effect on the temperature, atmospheric stabilities, and convective activity (Nowack et al., 2015). All these factors would impact water vapor and cloud formation.

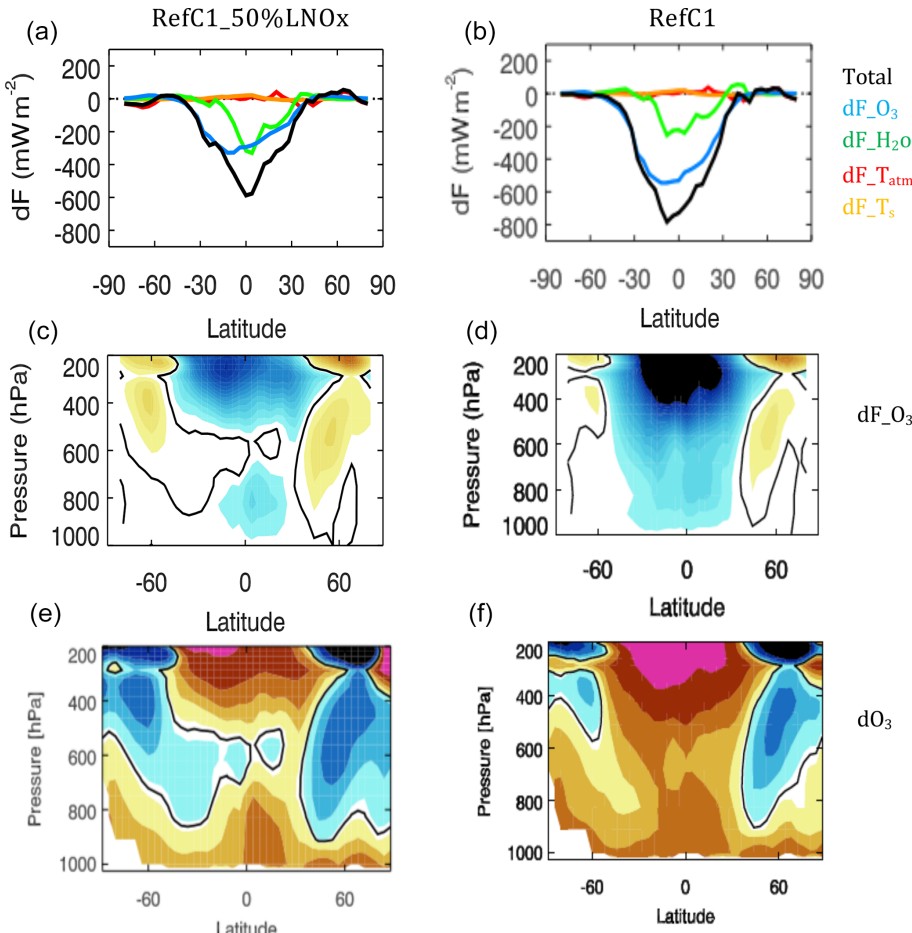

**Figure A1.** The comparison of the MRI-ESM1r1 experiment of half LNO$_x$ in total flux bias **(a, b)**, O$_3$ radiative bias **(c, d)**, and O$_3$ bias **(e, f)**. **(a, c, e)** New run with half LNO$_x$. **(b, d, f)** RefC1 run.

*Data availability.* The CCMI models' output were obtained from https://blogs.reading.ac.uk/ccmi/badc-data-access/ (last access: 30 December 2019).

The tropospheric chemical reanalysis data are available at https: //tes.jpl.nasa.gov/chemical-reanalysis/ (last access: 30 December 2019).

AURA TES IRK products can be download at https://tes.jpl.nasa. gov/data (last access: TS2).

*Author contributions.* LK and KWB designed the analysis and organized the paper. LK developed the IRK product, performed all the analysis, and drafted the paper. KWB helped interpret the results. HW and SK provided the code used to further develop the new IRK products. KM contributed the TCR-1 data. The other authors ran the individual model, contributed the model output, and helped revise the paper.

*Competing interests.* The authors declare that they have no conflict of interest.

*Special issue statement.* This article is part of the special issue "Chemistry-Climate Modelling Initiative (CCMI) (ACP/AMT/ESSD/GMD inter-journal SI)". It is not associated with a conference.

*Acknowledgements.* This work was conducted at Jet Propulsion Laboratory. Le Kuai and Kevin W. Bowman's research was carried out at the Jet Propulsion Laboratory, California Institute of Technology, under a contract with the National Aeronautics and Space Administration. Le Kuai and Kevin W. Bowman were supported under NASA ROSES NNH13ZDA001N-AURAST. The TCR-1 work was supported through JSPS KAKENHI grant numbers 26287117 and 18H01285 and by the Environment Research and Technology Development Fund (2-1803) of the Ministry of the Environment, Japan. The Earth Simulator was used to conduct chemical reanalysis calculations under the JAMSTEC Proposed Project and Strategic Project with Special Support. The EMAC model simulations were performed at the German Climate Computing Centre (DKRZ) through support from the Bundesministerium für Bildung und Forschung (BMBF). DKRZ and its scientific steering committee are gratefully acknowledged for providing the high-performance computing (HPC) and data archiving resources for the consortial project ESCiMo (Earth System Chemistry integrated Modelling). The GEOSCCM is supported by the NASA MAP program and the high-performance computing resources were provided by the NASA Center for Climate Simulation (NCCS). Eugene Rozanov is partially supported by the Swiss National Science Foundation under grant 200020_182239 (POLE) and the information gained will be used to improve next versions of the CCM SOCOL.

*Financial support.* This research has been supported by the NASA ROSES (grant no. NNH13ZDA001N-AURAST), the JSPS KAKENHI (grant nos. 26287117 and 18H01285), and the Swiss National Science Foundation (POLE; grant no. 200020_182239).

*Review statement.* This paper was edited by Pedro Jimenez-Guerrero and reviewed by two anonymous referees.

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

**Remarks from the typesetter**