# Peer review of "Attribution of Chemistry-Climate Model Initiative (CCMI) ozone radiative flux bias from satellites"

_Atmospheric Chemistry and Physics, 2019_

## Referee Comment (RC1) · Anonymous Referee #1 · 25 Jun 2019

Kuai et al. applied observational derived instantaneous radiative kernels to diagnose the source of biases in 9.6 um ozone-band radiative flux in an ensemble of chemistry-climate models. The result of the work is useful to guide efforts to improve chemistry-climate models and reduce the uncertainty in ozone RF estimates. The paper is in general well written. I recommend it for publication with a few minor comments.

The paper considers only clear-sky conditions as stated in page 3 line 28. Effect of cloud is not studied. Mention this in the title and abstract to avoid confusion.

Some details about how IRKs are computed can be useful. For example, how you compute deltaL/deltaq term in eq. (2)? Does this computation relies on prior information about vertical profiles of ozone, water vapor, Ta, and Ts? If so, how do they compare to the reanalysis used in this study or if any biases will impact the values of IRKs?

[Figure]

Eq. 3. Should there be some kind of layer thickness operator in eq.3? Model layers with different thickness should be weighted differently.

Page 6 Section 3: If I understand correctly, the model radiative flux bias is derived from eq (3) based on IRKs and the biases in O3, H2O, Ta and Ts. So how do these simulated ozone-band fluxes directly compare with satellite observations, e.g., figure 1?

Page 8 Line 10: complements

Page 9 Line 42: "while the absolute Ts bias has a larger than the Ts impact for most model". The sentence reads awkward and the second Ts should be Ta?

Page 10 Line 7-8: "negative global mean bias" and "biased low" are repetitive. Page 10 Line 21: the second Ta should be Ts? Page 14 Line 25-27,33-35: I do not understand the argument to explain the anticorrelation between o3 and OLR and no correlation between H2O and OLR.

---

## Author Comment (AC1) · 17 Jul 2019

We would like to thank the Editorial Review Board and the reviewers for their helpful comments. These feedbacks help to improve the clarity and readability of the manuscript. Please find our replies following the comments.

Anonymous Referee #1

Kuai et al. applied observational derived instantaneous radiative kernels to diagnose the source of biases in 9.6 um ozone-band radiative flux in an ensemble of chemistry-climate models. The result of the work is useful to guide efforts to improve chemistry-climate models and reduce the uncertainty in ozone RF estimates. The paper is in general well written. **I recommend it for publication with a few minor comments. The paper considers only clear-sky conditions as stated in page 3 line 28. Effect of cloud is not studied. Mention this in the title and abstract to avoid confusion.**

We have added a statement about clear-sky in the abstract in order to alert the readers of this important caveat. However, we prefer to keep the title short and focused.

Updated in abstract, page 1 line 41:
'To that end, we constructed observational instantaneous radiative kernels (IRKs) under clear-sky conditions, representing the sensitivities of the TOA flux in the 9.6-µm ozone band to the vertical distribution of geophysical variables, including $O_3$, $H_2O$, $T_a$, and $T_s$ based upon the Aura Tropospheric Emission Spectrometer (TES) measurements.'

(1) Some details about how IRKs are computed can be useful. For example, how you compute deltaL/deltaq term in eq. (2)? (2) Does this computation relies on prior information about vertical profiles of ozone, water vapor, Ta, and Ts? (3) If so, how do they compare to the reanalysis used in this study or if any biases will impact the values of IRKs?

(1) deltaL/deltaq term in eq. (2) is the analytic spectral radiance Jacobians calculated by TES radiative transfer model. A statement is added in the page 5 line 26.
'The partial derivative term is the spectral radiance Jacobians calculated analytically by TES radiative transfer model (Clough et al., 2006).'

S. Clough, M. Shepard, J. R. Worden, P. D. Brown, H. M. Worden, M. Lou, C. Rodgers, C. Rinsland, A. Goldman, L. Brown, A. Eldering, S. S. Kulawik, K. Cady-Pereira, G. Osterman, and R. Beer. Forward model and Jacobians for Tropospheric Emission Spectrometer retrievals. IEEE Transactions on Geoscience and Remote Sensing, 44(5):1308–1323, May 2006.

(2) No, these Jacobians do not rely on prior profile but rely on retrieved profiles since the IRK algorithm are developed from the TES operational retrieval algorithm. The retrieved atmosphere state ensure the sensitivity of the radiance to the variables are calculated for radiances with the best fit of the observed radiances. The reanalysis atmosphere doesn't keep the radiances with the best fit of the observed radiance.

We use the reanalysis data for the bias of the physical quantities in models for the sampling density purpose. The reanalysis ozone data is the assimilation data of TES ozone with regular grid of high spatial resolution. For the places with good TES observations, it is most approach TES data. For the places without enough TES observations, it rely on the nearby observations. This study was repeated with the model bias computed with the TES observations, the conclusion doesn't change by using two difference data as reference.

Eq. 3. Should there be some kind of layer thickness operator in eq.3? Model layers with different thickness should be weighted differently.

Yes, the layer thickness need to be considered. I included the layer thickness and its integration in equation (5), represented in variable 'l', page 7 line 5:

$$\delta F_q^j = \frac{1}{N_j} \sum_{i \in D_j} \sum_{l \in L} w_i \frac{\partial F_{TOA}^i}{\partial q^{i,l}} (q_{mod}^{i,l} - q_{assim}^{i,l}) \qquad (5)$$

where $w_i$ is area-weighted for the latitude bands, $D_j$ is a set of observed locations, $N_j$ is the number of locations in the domain of $D_j$ and tropospheric levels $L$ up to the tropopause.

Page 6 Section 3: If I understand correctly, the model radiative flux bias is derived from eq (3) based on IRKs and the biases in O3, H2O, Ta and Ts. So how do these simulated ozone-band fluxes directly compare with satellite observations, e.g., figure 1?

The CCMI modelling groups did not upload their model simulated ozone-band fluxes. They did, however, provide the total OLR (broad band flux). So we could not make such a comparison.

Consequently, we examined the correlation between the ozone-band flux bias to the OLR bias in Section 6.

Page 8 Line 10: complements

Corrected.

Page 9 Line 42: "while the absolute Ts bias has a larger than the Ts impact for most model". The sentence reads awkward and the second Ts should be Ta?

We rewrote the sentences as follows:
'$T_s$ bias is also meridionally weak relative to the flux bias in $O_3$ and $H_2O$ (Fig 4). With the exception of CMAM, the Ts ensemble global mean bias is less than 35 mWm$^{-2}$ (see Table 2). Figure 4 suggests the strong bias from Ts in CMAM ($-100.2$ mWm-2 ) comes from the two subtropical regions.'

Page 10 Line 7-8: "negative global mean bias" and "biased low" are repetitive.

We removed 'biased low'
'However, all the other models have a strong negative global mean bias and are mostly driven by the two major components ($O_3$ and $H_2O$).'

Page 10 Line 21: the second Ta should be Ts?

Page 10 line 21 the second Ta has been corrected to Ts.
'To further investigate, we investigated the vertically-resolved flux bias for $O_3$, $H_2O$ and $T_a$ (Fig. 5-7) and the global distribution for $T_s$ (Fig. 8).'

Page 14 Line 25-27,33-35: I do not understand the argument to explain the anticorrelation between o3 and OLR and no correlation between H2O and OLR.

We agree that the reasons are speculative because we do not have full access to the climate model RT code.  However, the O3 band is a part of the OLR band. So, if that bias increases, then it must be compensated elsewhere in order to maintain the same OLR. The lack of correlation between H2O band and OLR is less clear.  Further investigation is necessary to understand these correlations.

---

## Referee Comment (RC2) · Anonymous Referee #2 · 28 Jul 2019

Review of Kuai et al, "Attribution of Chemistry-Climate Model Initiative (CCMI) ozone radiative flux bias from satellites"

In this paper, Kuai et al. quantify and attempt to interpret model biases in outgoing longwave radiation (OLR) in the wavelength band of strong absorption by tropospheric ozone, 9.6 microns. Using assimilated data, they first construct instantaneous radiative kernels (IRKs) to quantify the sensitivity of OLR at 9.6 μ to four driving variables: tropospheric ozone, water vapor, atmospheric temperature, and surface temperature. The IRKs are calculated at a range of altitudes and across the globe. They then apply the observation-based IRKs to the biases in the four variables in an ensemble of climate models; this step yields the contributions of these variables to biases in OLR at 9.6 μ in each models. The subsequent analysis shows the importance of accurate representations of tropospheric ozone and water vapor in calculating OLR in this wavelength region.

The topic of the paper is important. Understanding the contribution of tropospheric ozone to current and future climate change requires models that can accurately simulate OLR in the relevant wavelengths. The calculations in the paper are fairly straightforward and yield some interesting results – e.g., that the overestimate of tropical tropospheric water vapor present in many models leads to an underestimate of OLR in the ozone wavelength band. But the paper feels like an early draft: much of the analysis in the paper is shallow and the paper is not well written.

I recommend acceptance only after major revisions.

**Major comments.**

1. A key reason to improve OLR at 9.6 μ is to improve estimates of the radiative forcing from the change in tropospheric ozone from the preindustrial era to the present-day or from the present-day to the future. The manuscript leaves the reader wondering how important are the biases uncovered in the manuscript. Do the biases have much of an effect on ozone forcing estimates over time?

2. In discussing the sources of biases in the four driving variables, the manuscript provides some detail about how modelers have struggled to improve estimates of tropospheric ozone (e.g., Oman et al., 2011 and 2013; Stenke et al., 2013; Revell et al., 2015). But nothing is said about model biases in the other three variables. Are these biases well-known or new to the community? If known, what are the potential reasons for these biases?

3. The paper focuses on OLR under clear-sky conditions, which is fine. But cloud cover varies significantly among models. Could variation in clouds also affect model estimates of ozone radiative flux – i.e., OLR at 9.6 μ? The text should include a discussion of how the focus on clear-sky conditions affects the conclusions.

4. The paper is poorly written with many lapses in English, about 1-3 per paragraph throughout. Those co-authors for whom English is a first language should read the manuscript much more closely and rewrite as needed. As is, the manuscript gives the impression that the co-authors did not read it carefully.

**Minor comments.**

Page 2. Line 10. Abstract should make clear that the overestimate of atmospheric opacity in the models is due to too much tropospheric ozone and/or water vapor. Also, the large number of significant digits looks suspect.

Page 2. Line 22. The text should make clear whether these estimates of radiative forcing are instantaneous or adjusted.

Page 3. Line 18. Text should clarify why the western Pacific is particularly opaque.

Page 6. Line 38. Reader wonders whether surface characteristics matter to OLR calculations, or are these captures by $T_s$?

Page 9. Line 41. "The flux bias due to Ta is found to be negligible in all models, which indicates that the Ta is relatively accurate." The statement should probably be qualified to say simply that the modeled Ta estimates provide reasonable radiative fluxes at 9.6 μ. The temperatures may not be so accurate for other purposes.

Page 10. Lines 20-25. The authors might consider reordering the figures so that the figures showing biases in each driving variable come right after the figures showing the biases in the fluxes attributable to that variable – e.g., Figure 9 right after Figure 5. That way readers can more easily see how the driving variables affect fluxes.

Page 11. Lines 20-25. Are these ozone biases, reported in ppb, weighted by air density? If not, perhaps they should be, given that the opacity of the atmosphere is affected by total ozone column, not average ozone mixing ratio in that column.

Page 12. Line 8. What updates did Oman 2011 and Oman 2013 implement in GEOSCCM to improve the model's representation of ozone? Readers will want to know.

Page 12. Line 25. The text should provide a reference to substantiate the claim that MRI-EMS1r1 overestimates lightning NOx emissions and underestimates convective updrafts in the tropics.

Page 12. Line 35. This reader does not understand "fact" 3, which gives this explanation for the bias in the radiative flux: "the systematic bias throughout the tropical troposphere, when propagate into the TOA flux, causes an accumulated bias in the radiative effect."

Page 13. Line 20. Why do the Ta biases in the tropics "shift between positive and negative vertically" in most models?

Page 14. Lines 6-35. The text should clarify what of value is learned in the analysis of how biases in 9.6 μ band affect biases in the entire OLR band.

Pages 14-16. The conclusion should emphasize that the paper focuses only on clear-sky OLR.

Figure 4. The caption could say that the black curves are the same as the colored curves in Figure 3.

Figure 5. What do the black curves represent?

Figure 13. Equations should be shown only for those correlations and slopes that are statistically significant.

Table 3. Are the mixing ratios for ozone and water weighted by air density in each layer?

---

## Author Response (AR1)

We would like to thank the Editorial Review Board and the reviewers for their helpful comments. These feedbacks help us to improve the clarity and readability of the manuscript. Please find our replies following the comments.

Anonymous Referee #2

Review of Kuai et al, "Attribution of Chemistry-Climate Model Initiative (CCMI) ozone radiative flux bias from satellites"

In this paper, Kuai et al. quantify and attempt to interpret model biases in outgoing longwave radiation (OLR) in the wavelength band of strong absorption by tropospheric ozone, 9.6 microns. Using assimilated data, they first construct instantaneous radiative kernels (IRKs) to quantify the sensitivity of OLR at 9.6 μ to four driving variables: tropospheric ozone, water vapor, atmospheric temperature, and surface temperature. The IRKs are calculated at a range of altitudes and across the globe. They then apply the observation-based IRKs to the biases in the four variables in an ensemble of climate models; this step yields the contributions of these variables to biases in OLR at 9.6 μ in each models. The subsequent analysis shows the importance of accurate representations of tropospheric ozone and water vapor in calculating OLR in this wavelength region.

The topic of the paper is important. Understanding the contribution of tropospheric ozone to current and future climate change requires models that can accurately simulate OLR in the relevant wavelengths. The calculations in the paper are fairly straightforward and yield some interesting results – e.g., that the overestimate of tropical tropospheric water vapor present in many models leads to an underestimate of OLR in the ozone wavelength band. But the paper feels like an early draft: much of the analysis in the paper is shallow and the paper is not well written.

I recommend acceptance only after major revisions.
Major comments.
1. A key reason to improve OLR at 9.6 μ is to improve estimates of the radiative forcing from the change in tropospheric ozone from the preindustrial era to the present-day or from the present-day to the future. The manuscript leaves the reader wondering how important are the biases uncovered in the manuscript. Do the biases have much of an effect on ozone forcing estimates over time?

We agree these are all very good points. Yes, the biases we studies have effect on the ozone forcing. We have showed in our early study (Kuai et al., 2017) that the tropospheric ozone radiative effect will be weaker under optical thick sky, such as with cold and wet condition through the secondary effect on ozone IRK. Therefore, the incorrect water vapor or temperature

will cause the response in IRK in 9.6 μm band and also effect the other band in OLR. And also the wrong water vapor and temperature will also cause incorrect cloud cover and then ozone loss. Too much or too less ozone and water vapor will have wrong heating rate and effect temperature and cloud. All these will effect the ozone forcing estimates from present day to the past.

In meanwhile don't have the O$_3$ RF estimation over time from CCMI models to do the similar analysis as Bowman et al., 2013 for ACCMIP. We have raised our request in the most recent CCMI workshop for the RF calculation from the CCMI models. Hope in near future, the data will be available. Then we will do more analysis to understand the uncovered biases and their effect on the ozone forcing over time.

We mentioned in the introduction how 9.6 m band bias would impact the ozone RF over time at Page 3, line 44.

'Aghedo, et al. (2011) applied the TES IRKs to evaluate the O3 radiative effect of chemistry-climate models' O3 biases in the Atmospheric Chemistry Climate Model Inter-comparison project (ACCMIP) (Lamarque, et al., 2013). Bowman et al. (2013) found model OLR bias due to O3 is correlated with RF in the ACCMIP models. This correlation helped to reduce the inter-model divergence in RF by about 30% (Myhre, et al., 2013).'

2. In discussing the sources of biases in the four driving variables, the manuscript provides some detail about how modelers have struggled to improve estimates of tropospheric ozone (e.g., Oman et al., 2011 and 2013; Stenke et al., 2013; Revell et al., 2015). But nothing is said about model biases in the other three variables. Are these biases well-known or new to the community? If known, what are the potential reasons for these biases?

The different biases in the models are known since there are regularly evaluation versus all kinds of observations but the reason are poorly understood. However, the biases in the four driving variables for CCMI study and their radiative biases are new to CCMI community and the reason are not fully understood. To my knowledge, for the CCMI comparison, there are not yet specific analysis about the lower tropospheric water vapor biases and temperature biases. Most of such studies are about past inter-comparison project, like CMIP6 or ACCMIP. For example, Lamarque et al., 2013 did a fairly rough analysis in water vapor biases for ACCMIP (Fig. S4). There is some information on the bias for the GEOSCCM in the technical report by Molod et al., 2012 (see figures 18 and 19). However, this is not the CCMI run specifically. Monks et al., 2015 showed the correlation between the intermodal differences in OH in the POLMIP study with inter-model differences in water vapor. They found GEOSCCM simulated specific humidity has high bias relative to MERRA reanalysis in most troposphere and also in mid troposphere compared to Atmospheric Infrared sounder (AIRS) data. Strode et al., 2015 section 3.4.3 discussed some water vapor biases impact on OH, methane and mehyl choloroform lifetime in GESCCM. Again these analyses are not for CCMI study and didn't provide any potential reasons for the biases.

We add some words at Page 15 line 38 as below.

'We compute the model biases against reanalysis data for four key variables: O3, H2O, Ta, and Ts. Especially for O3 biases, the newly developed TCR-1 O3 assimilation data (Miyazaki et al., 2015; Miyazaki and Bowman 2017) are, for the first time, used as the state-of-the-art benchmark for tropospheric O3 in models. These specific bias comparisons for the CCMI study cause the modelers to investigate the reasons for these biases and motivate them to improve their simulations. For example, MRI-ESM1r1 shows the reduced LNOx emission help to improve their tropical upper tropospheric O3 and its radiative bias.'

References mentioned above:
Lamarque, J.-F., Shindell, D. T., Josse, B., Young, P. J., Cionni, I., Eyring, V., Bergmann, D., Cameron-Smith, P., Collins, W. J., Doherty, R., Dalsoren, S., Faluvegi, G., Folberth, G., Ghan, S. J., Horowitz, L. W., Lee, Y. H., MacKenzie, I. A., Nagashima, T., Naik, V., Plummer, D., Righi, M., Rumbold, S. T., Schulz, M., Skeie, R. B., Stevenson, D. S., Strode, S., Sudo, K., Szopa, S., Voulgarakis, A., and Zeng, G.: The Atmospheric Chemistry and Climate Model Intercomparison Project (ACCMIP): overview and description of models, simulations and climate diagnostics, Geosci. Model Dev., 6, 179–206, https://doi.org/10.5194/gmd-6-179-2013, 2013.

Molod, A., Takacs, L., Suarez, M., Bacmeister, J., Song, I.-S., and Eichmann, A.: The GEOS-5 atmospheric general circulation model: Mean climate and development from MERRA to Fortuna, NASA, Goddard Space Flight Center, Greenbelt, MD, 2012.

Monks, S. A., Arnold, S. R., Emmons, L. K., Law, K. S., Turquety, S., Duncan, B. N., Flemming, J., Huijnen, V., Tilmes, S., Langner, J., Mao, J., Long, Y., Thomas, J. L., Steenrod, S. D., Raut, J. C., Wilson, C., Chipperfield, M. P., Diskin, G. S., Weinheimer, A., Schlager, H., and Ancellet, G.: Multi-model study of chemical and physical controls on transport of anthropogenic and biomass burning pollution to the Arctic, Atmos. Chem. Phys., 15, 3575–3603, doi:10.5194/acp-15-3575-2015, 2015.

Strode, S. A., Duncan, B. N., Yegorova, E. A., Kouatchou, J., Ziemke, J. R., and Douglass, A. R.: Implications of carbon monoxide bias for methane lifetime and atmospheric composition in chemistry climate models, Atmos. Chem. Phys., 15, 11789–11805, https://doi.org/10.5194/acp-15-11789-2015, 2015.

3. The paper focuses on OLR under clear-sky conditions, which is fine. But cloud cover varies significantly among models. Could variation in clouds also affect model estimates of ozone radiative flux – i.e., OLR at 9.6 µ? The text should include a discussion of how the focus on clear sky conditions affects the conclusions.

We added some discussion based on reviewer's feedback at Page 3, Line 26:

'In addition, the presence of clouds is the primary control on atmospheric opacity. Under the cloudy sky conditions, the roles of these variables other than cloud on TOA flux are much weaker. In addition, the variation in clouds could affect model estimates not only of the ozone but also of the flux sensitivity to ozone and other variables. Both ozone and sensitivity will impact the ozone radiative flux but in opposite directions. With cloud cover, the O3 loss will be reduced. That means too much cloud would lead to more ozone production. The presence of the

cloud would also cause weaker flux sensitivity to O3 and other variables (IRKs). Therefore, the cloud effect is a battle between the impact on ozone estimation and the radiative sensitivity to ozone (IRK). The differences in cloud variations between the models will complicate the radiative effect. Furthermore, the study of the cloud effect is also currently limited by the global observations of total cloud cover and IRK product under realistic cloud conditions. Without knowing which models have better cloud cover, we benefit from using IRK based on the observed cloud free data by TES. Therefore, here we first try to access the role of O3, H2O, Ta and Ts in the variation of the TOA flux without cloud effect.'

4. The paper is poorly written with many lapses in English, about 1-3 per paragraph throughout. Those co-authors for whom English is a first language should read the manuscript much more closely and rewrite as needed. As is, the manuscript gives the impression that the co-authors did not read it carefully.

We have got the professional grammar check after we addressed all comments from two reviewers.

Minor comments.

Page 2. Line 10. Abstract should make clear that the overestimate of atmospheric opacity in the models is due to too much tropospheric ozone and/or water vapor. Also, the large number of significant digits looks suspect.

We addressed reviewer's suggestions by adding words for the reason of the opacity in Page 2, line 8 as below and change the number form 132.9 to 133.

"Overall, the multi-model ensemble mean bias is –133 ± 98 mWm-2, indicating that they are too atmospherically opaque due to trapping too much radiation in the atmosphere by overestimated tropical tropospheric O3 and H2O. Having too much O3 and H2O in the troposphere would have different impacts on the sensitivity of TOA flux to O3 and these competing effects add more uncertainties on the ozone radiative forcing."

Page 2. Line 22. The text should make clear whether these estimates of radiative forcing are instantaneous or adjusted.

This is adjusted radiative forcing. Page 2, Line 21 has been updated as

'Tropospheric O3 adjusted RF ranges widely from +0.2 to +0.6 Wm-2 computed from chemistry-climate model ensembles (IPCC AR5, 2013) (Bowman, et al., 2013;Stevenson, et al., 2013).'

Page 3. Line 18. Text should clarify why the western Pacific is particularly opaque.

Page3, line 15 has been updated:

"For example, O3 changes in more opaque regions, e.g., the Western Pacific, a wet region due to convection, result in a much smaller change in TOA flux than in more transparent regions, e.g., the Middle East, a dry region due to downwelling (Kuai, et al., 2017)."

Page 6. Line 38. Reader wonders whether surface characteristics matter to OLR calculations, or are these captures by Ts?

Yes, surface characteristics like emissivity will effect the OLR calculations. For different surface types, like rock or vegetation, the emissivity could vary from 0.9 to 1. The surface type could relate to the differences of Ts but such difference would be much smaller than seasonal variation of Ts. Ts is a dominate factor to determine OLR. Here we are discussing the variability of the IRK for $T_s$ is mainly controlled by $T_s$.

Page 9. Line 41. "The flux bias due to Ta is found to be negligible in all models, which indicates that the Ta is relatively accurate." The statement should probably be qualified to say simply that the modeled Ta estimates provide reasonable radiative fluxes at 9.6 μm. The temperatures may not be so accurate for other purposes.

We addressed the comments now at Page 10, line 5.
'The flux bias due to Ta is found to be negligible in all models, which indicates that the model Ta estimates provide reasonable radiative fluxes.'

Page 10. Lines 20-25. The authors might consider reordering the figures so that the figures showing biases in each driving variable come right after the figures showing the biases in the fluxes attributable to that variable – e.g., Figure 9 right after Figure 5. That way readers can more easily see how the driving variables affect fluxes.

We did reconsider the reviewer's great suggestion but decided to keep the current structure. Section '5.3 Vertically-resolved radiative bias of the $O_3$, $H_2O$ and T' introduce the partitioned radiative biases for all four variables and summarized their features briefly. This let the readers to do the cross comparison between the variables. Then, in the section '5.4 The Spatial Source of TOA Flux Bias', we did the discussion in the style as reviewer's way. For example for $O_3$, we talk about Fig. 5 and Fig. 9. For $H_2O$, we discuss Fig. 6 and Fig. 10 together.

Page 11. Lines 20-25. Are these ozone biases, reported in ppb, weighted by air density? If not, perhaps they should be, given that the opacity of the atmosphere is affected by total ozone column, not average ozone mixing ratio in that column.

Yes, the ozone biases are reported in ppb. When we do the vertical integration in free troposphere (Eq. 5), the column density is considered.

Page 12. Line 8. What updates did Oman 2011 and Oman 2013 implement in GEOSCCM to improve the model's representation of ozone? Readers will want to know.

We added more detailed discussion about how does GEOSCCM improved their ozone representation in Page 12, line 26.

'The GEOSCCM has been used to study the tropospheric O3 response to variations in the El Nino-Southern Oscillation (ENSO) where Oman, et al., (2011;2013) compared the model to satellite observations. These regular comparisons may have led to the improved simulation of tropospheric O3 profiles and consequently lower vertical O3 bias. The GEOSCCM model in the CCMI study uses the tropospheric/stratospheric chemical package developed within the Global Modeling Initiative (GMI) program (Duncan, et al., 2007) which has more realistic ozone chemistry, an internally generated quasi-biennial oscillation, an improved air/sea roughness parameterization and other improvements (Oman et al., 2014).

Nielsen et al., (2017) showed that GEOSCCM successfully reproduces the changes in the quasi-global (60°S–60°N) annual-mean trend in total O3 column since 1960s to the present day. For the present-day atmosphere, simulated tropospheric partial column O3 from GESCCM Ref-C1 for CCMI was compared to satellite observations of OMI and MLS (Ziemke et al., 2011). The differences are mostly a few Browner Dobson (DU) except the Northern Hemisphere subtropics and middle latitudes in autumn and winter with the 4–6 DU bias which are under investigation.'

New References:
Nielsen, J. E., S. Pawson, A. Molod, B. Auer, A. M da Silva, A. R. Douglass, B. Duncan, Q. Liang, M. Manyin, L. D. Oman, W. Putman, S. E. Strahan and K. Wargan, 2017. **Chemical mechanisms and their applications in the Goddard Earth Observing System (GEOS) earth system model**. Journal of Advances in Modeling Earth Systems, 9, 3019-3044, DOI: https://doi.org/10.1002/2017MS001011.

Molod, A., Takacs, L., Suarez, M., Bacmeister, J., Song, I.-S., & Eichmann, A. (2012). *The GEOS-5 Atmospheric General Circulation Model: Mean Climate and Development from MERRA to Fortuna* (NASA Tech. Rep. Series on Global Modeling and Data Assimilation, NASA/TM-2012–104606, Vol. 28, 117 pp.). Greenbelt, MD: NASA Goddard Space Flight Center.

[revised manuscript text omitted]

Page 12. Line 35. This reader does not understand "fact" 3, which gives this explanation for the bias in the radiative flux: "the systematic bias throughout the tropical troposphere, when propagate into the TOA flux, causes an accumulated bias in the radiative effect."

We revised the statement now at Page 13 Line 19 as below

'3) the simulations with the systematic bias throughout the tropical troposphere, when vertically integrated, accumulated into a larger column bias when compared to the models with vertically random biases.'

Page 13. Line 20. Why do the Ta biases in the tropics "shift between positive and negative vertically" in most models?

That means simulated air temperatures stay around the reanalysis profiles. These models better represent the air temperature than trace gases like $H_2O$ and $O_3$.
We added the sentences in now Page 14, Line 7:

'The oscillated $T_a$ biases suggest that simulated air temperatures stay around the reanalyzed profiles. These models better represent the air temperature than the trace gases like $H_2O$ and $O_3$.'

Page 14. Lines 6-35. The text should clarify what of value is learned in the analysis of how biases in 9.6 µ band affect biases in the entire OLR band.

We added some statements for the value of the understanding about how biases in 9.6 µ band affect biases in the entire OLR band. Now Page 15, line 31:

'The anti-correlation between the biases in the 9.6-µm band and in the entire OLR band would suggest some bias drivers in the 9.6-µm band must play different roles at the other part of the OLR band. The further investigation of these processes would help to explain the radiative effect of different biases on the OLR estimations from models (Huang et al., 2008; Huang et al., 2014).'

New References:
Huang, X. L., W. Yang, N. G. Loeb, and V. Ramaswamy (2008), Spectrally resolved fluxes derived from collocated AIRS and CERES measurements and their application in model evaluation: Clear sky over the tropical oceans, J. Geophys. Res., 113, D09110, doi:10.1029/2007JD009219.

Huang, X. L., Chen, X. H., Potter, G. L., Oreopoulos, L., Cole, J. N. S., Lee, D. M., & Loeb, N. G. (2014). A global climatology of outgoing longwave spectral cloud radiative effect and associated effective cloud properties. Journal of Climate, 27(19), 7475–7492. https://doi.org/10.1175/JCLI-D-13-00663.1

Pages 14-16. The conclusion should emphasize that the paper focuses only on clear-sky OLR.

We added the statement for the clear sky in the following sentence. Now Page 15, line 31

'We have demonstrated a new method to quantitatively attribute the biases in $O_3$ band TOA flux from chemistry-climate model ensembles to $O_3$, $H_2O$, $T_a$, and $T_s$ radiative components without cloud effect using observationally-constrained IRKs in the clear sky.'

Figure 4. The caption could say that the black curves are the same as the colored curves in Figure 3.
Added to the caption.
'Figure 4. The attribution the total TOA flux bias for each model to four dominant components and their latitudinal distribution. The black curves are the same as the colored curves in Figure 3.'

Figure 5. What do the black curves represent?

We add the words as below
'Figure 5. Vertical resolved $O_3$ radiative bias. The black curves are the zero lines.
 Figure 6. Vertical resolved H2O radiative bias. The black curves are the zero lines.
 Figure 7. Vertical resolved $T_a$ radiative bias. The black curves are the zero lines.
 Figure 9. The zonal averaged vertical-latitudinal distribution of $O_3$ model biases to the TCR-1 $O_3$ assimilation data. The black curves are the zero lines.
 Figure 10. The zonal averaged vertical-latitudinal distribution of $H_2O$ biases (model to the ERA reanalysis data). The black curves are the zero lines.
 Figure 11. The zonal averaged vertical-latitudinal distribution of $T_a$ biases from models to the reanalysis data. The black curves in the contour are the zero lines.'

Figure 13. Equations should be shown only for those correlations and slopes that are statistically significant.

OK. We removed the equations of the fitting lines with weak correlations.

[Figure]

Figure 13. The correlation of the ozone band TOA flux biases to the model calculated broadband OLR (a) and the correlation of the attributed radiative components to the broadband OLR (b – e).

Table 3. Are the mixing ratios for ozone and water weighted by air density in each layer?

No. This is just the simple regional concentration average. We didn't compute the column bias by account for air density in each layer. The vertical weighting is considered while multiply with IRK and integral into tropospheric column.

[revised manuscript text omitted]